# Exploitation and exploration in text evolution. Quantifying planning and translation flows during writing

Donald Ruggiero Lo Sardo[1,2,3]*, Pietro Gravino[1,2,4], Christine Cuskley[5], Vittorio Loreto[1,2,4,6]

**1** Sony Computer Science Laboratories Rome, Joint Initiative CREF-SONY, Centro Ricerche Enrico Fermi, Rome, Italy, **2** Centro Ricerche Enrico Fermi, Rome, Italy, **3** Complexity Science Hub Vienna, Vienna, Austria, **4** Sony Computer Science Laboratories Paris, Paris, France, **5** Language Evolution, Acquisition and Development Group, Newcastle University, Newcastle Upon Tyne, United Kingdom, **6** Physics Department, Sapienza University of Rome, Rome, Italy

* donaldruggiero.losardo@sony.com

**Data Availability Statement:** Data cannot be shared publicly because of it contains sensitive information. Data are available from the SONY CSL Institutional Data Access (contact via pietro.gravino@sony.com and elisabetta.falivene@sony.

## Abstract

Writing is a complex process at the center of much of modern human activity. Despite appearing to be a linear process, writing conceals many highly non-linear processes. Previous research has focused on three phases of writing: planning, translation and transcription, and revision. While research has shown these are non-linear, they are often treated linearly when measured. Here, we introduce measures to detect and quantify subcycles of planning (exploration) and translation (exploitation) during the writing process. We apply these to a novel dataset that recorded the creation of a text in all its phases, from early attempts to the finishing touches on a final version. This dataset comes from a series of writing workshops in which, through innovative versioning software, we were able to record all the steps in the construction of a text. 61 junior researchers in science wrote a scientific essay intended for a general readership. We recorded each essay as a *writing cloud*, defined as a complex topological structure capturing the history of the essay itself. Through this unique dataset of writing clouds, we expose a representation of the writing process that quantifies its complexity and the writer's efforts throughout the draft and through time. Interestingly, this representation highlights the phases of "translation flow", where authors improve existing ideas, and exploration, where creative deviations appear as the writer returns to the planning phase. These turning points between translation and exploration become rarer as the writing process progresses and the author approaches the final version. Our results and the new measures introduced have the potential to foster the discussion about the non-linear nature of writing and support the development of tools that can lead to more creative and impactful writing processes.

## Introduction

In predominantly literate societies, the ability to read and write effectively becomes an essential capacity for individuals in the knowledge economy. Literacy is a limiting factor in measures of

com) for researchers who meet the criteria for access to confidential data.

**Funding:** SONY CSL Paris funded the work of Donald Ruggiero Lo Sardo, Pietro Gravino and Vittorio Loreto La Sapienza university of Rome funded the work of Donald Ruggiero Lo Sardo and Vittorio Loreto Christine Cuskley was funded in part by ESRC-SDAI grant ES/T005955/1 The funders had no role in study design, data collection and analysis, decision to publish, or preparation of the manuscript.

**Competing interests:** The authors have declared that no competing interests exist.

well-being, for example, self-assessed health, trust, income, and integration [1]. Anyone who sets out to communicate in writing is presented with many challenges, such as constructing a logical argument, weaving a comprehensible and intriguing narrative, and making the right lexical and grammatical choices. How do writers organize their thoughts into a narrative that others can read, understand, and possibly even enjoy? A body of theoretical and experimental work across psychology, linguistics, and education has developed several models of the process of writing. However, work looking at naturalistic data which documents not only the end product, but the *process* of writing, is lacking. In the current paper, we provide a framework for analyzing naturalistic writing in process in light of these models, using versioning software that documents the full process of writing. First, we introduce the broad strokes of how the process of writing is understood in cognitive science. We highlight that while phases of the writing process are acknowledged to be non-linear and feed back on one another, they are often treated as discrete entities methodologically. To remedy this, we introduce a method analyzing the entire process of creating a finished piece of writing focusing on signals arising from the text itself which indicate transitions between phases.

## Understanding writing

Cognitive scientists have identified three phases in the writing process that entail different sub-tasks: planning, translation and transcription (sentence generation), and revising [2]. The planning phase involves identifying the overall aims of a piece of writing, and the organization of initial ideas. The translation and transcription phase is where these ideas take concrete linguistic form, involving moving ideas to working memory (translation), activation of orthographic knowledge, and the engagement of motor areas involved in handwriting or typing (transcription). Finally, the revision stage involves evaluating and re-evaluating text and making changes. Crucially, this process is fundamentally non-linear because the revision phase entails repeated cycles of planning and translation: these phases are fundamentally interwoven [2]. The act of revision often involves adapting the initial plan, and enacting these adaptations as new sentence generation or alteration events. Put differently, these phases "are not strictly sequential; rather [they] interact recursively. . .can interrupt each other, and are embedded in each other" (p.2) [3].

Researchers in education and psychology have used a variety of approaches to study writing, with research often focusing on the process of learning to write during childhood [4]. This work has used three main approaches that give access to different kinds of information on the writing process: protocol analysis [5], task-driven experiments that are manually annotated [6], and keystroke logging [7]. Protocol analysis involves introspection on the process of writing from the author, either during the process of writing (concurrent think-aloud task) or after the writing event is complete (retrospective; [8]). In either case, the experience is recorded and scrutinized for evidence of the underlying process by which the writer has developed the text [2]. This kind of approach can play a key role in identifying cognitive processes involved in writing and *generating* hypotheses [9], but has limitations: it is not ideal for *testing* hypotheses, and difficult to scale up across large samples of writers.

Task-driven experiments are better suited to testing specific predictions and involve pre-specified limitations or constraints placed on the writing process. This may take the form of limiting or expanding the time available for particular phases or putting particular constraints on each phase. While this approach allows researchers to test specific hypotheses—for example, whether limiting or expanding time in the initial planning phase can affect text quality [6] —these constraints inherently disrupt how writers may naturally approach the task of writing.

Lastly, keystroke logging records each character typed (or removed) by a writer to investigate writing in real time. This approach has the advantage of being unobtrusive on its own, but is often necessarily combined with other methods and measures, including protocol analysis [10], eye-tracking [11], fMRI, galvanic skin response (GSR) or EEG (see [7] for an overview). However, on its own, keystroke logging can be *too* fine-grained a measure in isolation: "it is often difficult to connect the fine grain of logging data to the underlying cognitive processes" [12] (p.358). In other words, the focus on character-level edits can make it difficult to connect keystroke data to cognitive phases of writing, unless this data is considered in the context of other qualitative or quantitative measures. Where keystroke data is interpreted in isolation, it is often the *absence* of typing that draws attention: for example, pauses in keystrokes longer than about a second during otherwise continuous writing have been interpreted as indicative of discourse-level linguistic processing in chat contexts [13]. However, the specific evolution of a text over time at a higher word, sentence, or paragraph-level is generally not undertaken using this kind of data. In other words, keystroke logging in isolation is often used to analyze *when* writers focus their efforts, but not the temporal dynamics of *where* and *how* they allocate their focus in terms of, for example, the word or sentence level within a text.

## Investigating recursive sub-cycles

These methods converge on the fact that writing is not only a complex task but a fundamentally dynamic one. The writer starts by planning: forming a rough idea, or model, of what their final output will be [14]. However, this model must continuously adapt: the final version must conform to variations of its constituent parts, such as individual sentences or even words [15, 16]. For example, even a single lexical change to make terminology consistent throughout a piece of writing can cascade to larger structural changes to ensure this consistency is perceived as intentional by the reader. Other examples of adaptation points might be rhetorical solutions that do not satisfy the author or instances where, through the writing process, some unforeseen clarity on the content or new idea is found by the author that fundamentally reshapes the initial plan. These adaptation points are what shift the writer into repeated sub-cycles of planning and translation. In analogy with the study of strategies in evolutionary game theory we also use the term exploration to refer to planning phases, where the author tests different options for the draft.

Despite the well-acknowledged inter-relatedness of these phases, studies on the cognitive processes involved in writing tend to measure the different phases of writing discretely, even when they are all considered in a single study. Previous work, in particular, often treats planning as a distinct phase that begins and ends prior to the act of putting words to the page [6, 17–20]. Here, we focus on detecting shifts into and out of planning and translation during revision. To distinguish recursive sub-cycles of planning and translation from the overall cognitive model of planning, sentence production, and revision, we define reversions into planning as *exploration*. In other words, when writers revisit the planning phase during the revision process, they explore the shifting possibility space created by their own writing. When they choose where to implement intensive changes, they shift back into the translation and transcription phase. We operationalize each of these using new measures applied to a novel dataset that uses versioning software to track changes at a higher level than keystroke logging, on the order of minutes rather than seconds.

The ability to detect these shifts could provide the basis for important tools for improving writing: as the intervals between adaptation points take on particular patterns, we may be able to detect when a writer is reaching an optimization point for a text—or when continuing to push on the process may lead to diminishing returns. While natural language processing has

made strides in many areas of text analysis including general language comprehension [21], automatic translation [22], and text generation [23], author identification [24], topic modeling [25] and sentiment analysis [26], the *process* of effective writing has received little attention from more computationally focused perspectives. To the best of our knowledge, the study of the dynamic process of writing through the analysis of evolving versions of texts is limited, although some work has looked at naturally growing corpora such as tweets or newspaper publications [27, 28]. This is due, in part, to the fact that machine learning models are constrained by the data used for training, both in terms of the nature of the data and the large volume of it they generally require. Machine-learning approaches would require a large corpus of different draft versions documented and systematically annotated in terms of their evolution over time, but current training data generally only consists of finished texts. Data documenting the iteration and evolution of individual texts—especially at the scale required for many NLP approaches—is scarce, if not absent.

Here, we propose a new method that can efficiently detect patterns indicative of writing sub-cycles in much smaller datasets. This method presents a much lower overhead than previous work such as S-notation [16], given that it does not require keystroke logging to reconstruct the activity of the author. Since they rely on automatic versioning that is widespread across many text-editing tools, the measures introduced here can be used to investigate numerous text-based tasks for which data is already available, reducing the overhead in the data gathering process. This kind of data allows us to reconstruct a representation of the editing process which, in turn, allows us to measure its temporal complexity. This allows us to understand when an author is in an exploratory mindset, when they are effortlessly translating ideas onto the page, and when they are shifting rapidly between these tasks.

We present this novel analytical method using a new dataset composed of the diachronic series of versions of texts written in the framework of the first three editions of the *Scientific Evolutionary Writing* workshops (www.sew-workshop.org), described in detail below. We introduce three key measures of writing behavior based on simple Edit Distance (ED): complexity, exploration, and the twist ratio. The complexity measure shows that although texts are designed to be read linearly, the process of writing them is fundamentally non-linear. By representing editing events in a multi-dimensional space, we can observe how often and to what extent writers explore this space, and detect periods of "translation flow", when writers put words to the page with ease. Finally, we can contrast these two phases of exploration and "translation flow", detecting when writers seem to be "twisting" between exploration and what evolutionary theory calls exploitation.

## Data collection

The Scientific Evolutionary Writing (SEW) workshops are an ongoing series of classes on writing and editing scientific text intended for a general audience. Here, we report data from 61 participants across three SEW workshops which took place in 2019 and 2020, in Paris (Sony Computer Science Labs, N = 19), Vienna (Complexity Science Hub, N = 24), and Bern (Albert Einstein Center for Fundamental Physics, N = 18). Participants were recruited through online promotion and word of mouth from the academic community. Each workshop took place over 2 to 4 days, during which participants attended seminars on writing and editing strategies. Participants provided written informed consent and were all over the age of 18. More than 50% of participants had authored between 1 and 5 academic works (including unpublished submissions). The workshop was attended by academics (including Ph.D. students and early career researchers) with the most common research interests being in the fields of complexity science, computer science, and physics. Informed consent was required of participants.

Participants where asked to:—release the rights on the whole production at the workshop to Sony CSL, Sapienza and Inserm for scientific purposes only (they keep the intellectual property rights on my production);—allow the use of my personal data for scientific purposes only, provided that they are not disseminated other than in aggregate form. The participants were a diverse group, with a high fraction of science professionals (Ph.D. candidates to full professors) for whom English was not their mother tongue. Thus, the results should be evaluated in the broad context of L2 writers, but, as van Weijen et al. [29] have shown, the patterns of occurrence for cognitive activities during writing in L1 and L2 writers are alike, with L1 writers showing greater variance but similar behaviors on average to L2 writers. Moreover, our participants had significant previous experience writing in English given their academic backgrounds, and other aspects of the workshops were conducted in English.

During these workshops, participants were given writing advice from professional science writers, focusing on editing strategies and their similarities with the evolution of a biological system. The final goal was to write an essay at least six paragraphs long, intended to be a scientific report about the participant's work written for a general audience. The exercise was otherwise open-ended, with motivation coming directly from the participants, with the ultimate aim of using the final product for their work. Participants were given time to develop their work throughout the workshop to put the expert advice into practice. The writing phases were unsupervised, so other scheduled aspects of the workshops did not strongly impact workflow. The open-ended nature of the exercise, as well as the strong connection to participants' own work, was essential for the investigation of the natural writing process since motivation has a well-documented effect on text quality [30].

There were no exclusion criteria for the participants in the workshop itself, but the analysis only includes data from participants whose text changed at least 10 times over the course of the workshop, indicating active participation (numbers in Table 1 refer to active participants). Data on the demographic composition of the workshop participants was not systematically collected. With respect to the main metrics defined in this work, (process complexity, exploration, and twist ratio) we observe consistent distributions across the three workshops, indicating broad consistency across groups. For further details see Table 1.

In workshops 1 and 2, participants were asked to write their work using the Google Docs platform. In workshop 3, participants used the WeWrite platform, a tool developed by Sony CSL specifically to leverage the navigation of network-like structured texts to expose and manage the complexity of writing, writing collaboratively, and managing complex projects with multiple possible alternative texts. Both tools provide inbuilt versioning but with some minor differences. Google Docs creates a snapshot of the draft with uneven frequency, limiting the checkpoints to periods of activity. During these periods, the frequency was of $\sim 1$ checkpoint/minute. WeWrite instead allowed for a checkpoint every 3 minutes, irrespective of whether the writing activity was performed or not. To make all datasets consistent, versions that were identical to the previous one were excluded from the analysis.

**Table 1. A table detailing location, recruitment periods and number of participants for the scientific evolutionary workshops.**

|  | SEW 1 | SEW 2 | SEW 3 |
|---|---|---|---|
| Institute | SONY CSL | Complexity Science Hub | Albert Einstein Center for Fundamental Physics |
| Location | Paris | Vienna | Bern |
| Recruitment start | 22.01.2019 | 31.01.2019 | 17.12.2019 |
| Recruitment end | 30.01.2019 | 19.02.2019 | 14.01.2020 |
| Participants | 19 | 24 | 18 |

In the following sections we will describe the analysis of this data divided in three main contributions: a measure of the complexity of the writing process in terms the degree to which it is a sequential process, a measure of the amount of exploratory work that goes into the writing of a text but not into the final version and a measure of the translation flow to exploration ratio in the writing steps.

## Measuring complexity

To examine how structured the writing process is, we must first define the unit of text that best represents the activity of the writer we wish to capture. Like spoken language, written text can be analyzed at a variety of levels, by considering characters, words, sentences, paragraphs, or even larger units of structure (e.g., chapters in longer works). During the writing process, any kind of unit can be deleted, changed, and inserted. In short, any measurable unit of the text can be treated as an atomic unit of behavior for the writer. Different scales will register different kinds of writing activity: if we are interested in transcription (the process of encoding into orthographic units), for example, character-level analyses could detect this well. However, if we are more interested in translation (the process of encoding ideas into meaning-bearing linguistic units like words, phrases, and sentences), character analyses may be too granular and might obscure interesting patterns. For example, if a writer changes 7 characters, analysis exclusively at the character level might not detect this as a change to a lexical unit.

The right scale for a dataset is one for which the correlated (i.e., adjacent) edits emerge as one editing event. To determine the right scale, we computed the Edit Distance (ED) between two consecutive versions of a draft at the character, word, and paragraph levels. At each level, we annotate each segment, with 1 if it was changed and 0 otherwise. This results in a sequence of zeros and ones for each level, $\{B\}$, which characterizes the edit sequence at that level. Many adjacent edits will appear as a cluster of 1s within this sequence. However, many adjacent edits at, for example, the character level, are likely to be tied to a higher-level event (e.g., changing a word). Thus, to determine the level at which we can best detect related edits as a single event, we want clusters of 1s in $B$ to be minimal; in other words, edits in $B$ should be no more likely to be clustered than in a randomly reshuffled sequence of the same events.

To measure this, we adopted Bhattacharyya distance [31], which measures the similarity of two probability distributions. The level of granularity at which $B$ is minimally distant from a shuffled version of itself indicates a level where we are capturing distinct editing events. For our dataset, the Bhattacharyya distance is minimized at the sentence level, indicating that this is the resolution at which we can best detect atomic editing events. This will be the unit used in the Wagner–Fischer algorithm that ultimately provides the elements for the calculation of complexity ED [32]. The result that sentence-level edits are most relevant in our dataset is consistent with the finding that translation processes at the sentence level engage working memory [33].

Now, let us imagine you take a page of text, divide it into sentences, and look at it again after the author has worked on it for some time. Some sentences will have changed, some will have been deleted, and some will be entirely new. Much like a geneticist would do, we can match sentences that stayed the same and treat the changes as mutations. Comparing one version to the next, we can imagine they represent two paths through the same space. If they share common sentences, those are the places where the two versions match and the paths meet. The places where they do not match are where the author has intervened. These interventions may create a whole new section, change some parts the writer was unsatisfied with, and potentially affect the rest of the work. How can we represent this process in an informative way?

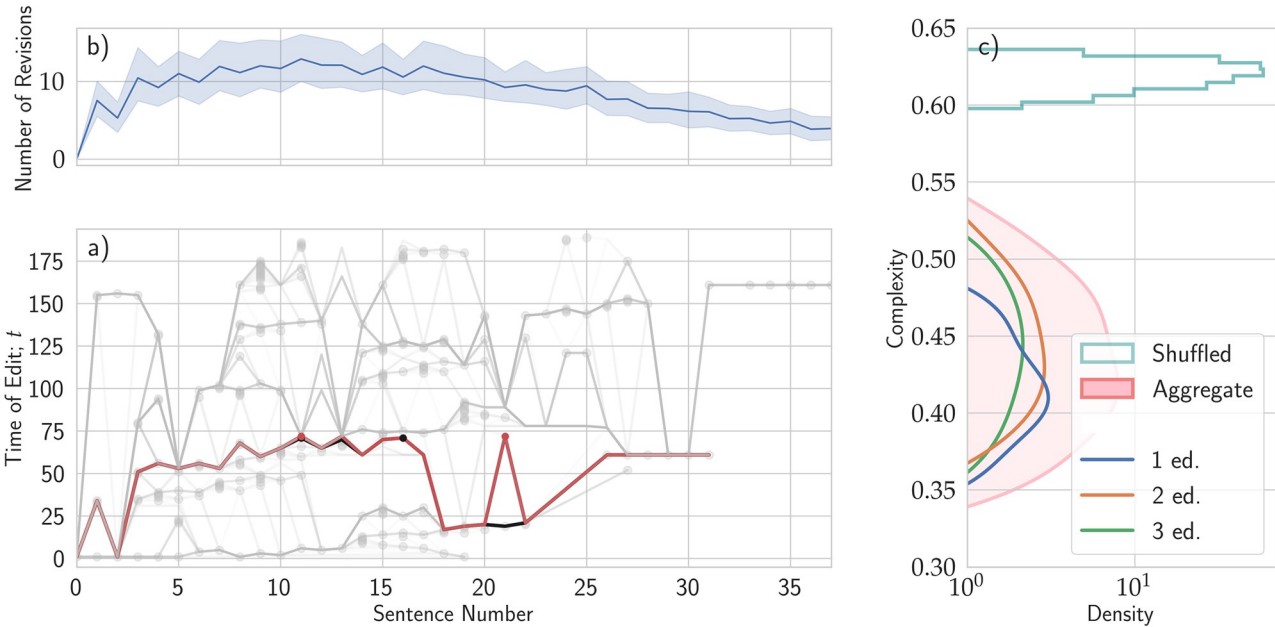

**Fig 1. Complexity of the writing process.** (a) The cloud of all versions of the text produced by a generic author. Each sentence written, even those deleted in the ongoing process, is represented as a point in this graph. The position on the horizontal axis represents the ordinal location of a sentence in a given draft version, the Sentence Number. The position on the vertical axis represents the version of the draft a sentence first appeared in, the Time of Edit $t$. A point in $\{x, t\}$ is a sentence first appearing as the $x$th sentence of the $t$th version of the draft. Focusing on a specific Time of Edit, the image shows all the sentences that have been revised at that time. Focusing on a particular Sentence Number, the image shows all the times at which that sentence has been edited. Each version of the draft is displayed by drawing a semi-transparent line through its ordered component sentences. Sequences of sentences preserved among different draft versions are more opaque. An example of two consecutive versions of the draft is highlighted in black [version 70] and red [version 71]. The 21st sentence of version 70, first appearing in version 20, has been edited in version 71. (b) We report the average number of edits of each sentence across all authors. The horizontal axis displays the position of each sentence in the text, while the vertical axis reports the number of edits. The blue area denotes the 99.5% confidence interval over the distribution using the bootstrapping procedure [$N$ = 1000]. (c) The distribution of complexity values. The pink histogram displays the values of the complexity of the writing processes recorded over all the workshops. Shaded areas are kernel density estimates for the individual workshops. The cyan line displays the probability density function computed over versions generated by randomly permuting the sequence of edits.

For each subsequent version of a draft after the first, we can define the edit sequence $\{B(t)\}$. In order to correctly compare different versions we add place-holder elements (empty strings) for the sentences that will be inserted later in the draft. At the same time we keep deleted sentences as empty place holder elements. This way, unmodified sentences will correspond to the same positional index as the draft evolves. Using this succession of sequences, we can build a cloud visualization of the writing process as reported in the (a) panel of Fig 1.

Equipped with the edit sequence we can investigate the structure of the process. To clarify, when we talk of structure in the context of complexity, we talk of the structure of edits as mapped onto the unidimensional space of the text, i.e. the x-axis of Fig 1(b). If the author edits in particular places more than others, the editing process is simple and predictable: if we had to predict where the writer would edit next, it is likely to be around a few focal points. On the other hand, if the edits are more evenly distributed across the text, they are the result of a more complex and less predictable process; if we had to predict where the writer would edit next, it might be anywhere. Note that here our dimension of interest is not temporal, but spatial: we are not looking at the dynamic process of writing over time, but by looking at a "flattened" array of changes defined by the end product, we obtain a measure representing where the writer's efforts were focused in the text. The spatial distribution of edits represents the distribution of the author's effort.

We quantify this using Shannon entropy [34]; texts where effort was spatially focused will have low entropy (converging to 0 where only one sentence was edited repeatedly), while texts with spatially distributed effort will have higher entropy. To compare different drafts that vary significantly in size, we must normalize by maximal entropy for a work of a specific size (number of sentences). This measure of complexity is referred to as the Shannon-Wiener diversity index and it is commonly used as a proxy for the complexity of an ecosystem [35].

## Results

In the bottom left panel of Fig 1, we see a synthetic representation as a cloud of the whole writing process. Each line of the cloud is a specific version of the text, organized along the *x*-axis in sentences, numbered from *zero* to the maximum number of sentences for each version. The first version is visible at the bottom as a straight line traversing the first version of each sentence from left to right. Later versions are jagged lines with vertices representing newer versions of the sentences. The further up a sentence is shown, the later it was written. The jagged line above all the others represents the last version. Lines are semi-transparent to highlight the parts common to several versions. The darker the line, the more versions share that portion of the text.

We can see that the author occupies much of the space of the figure. This evidence implies that work is neither sequential nor spatially localized. If the author had mainly intervened on a specific part of the text, the corresponding cloud of versions would be akin to a horizontal line apart from one section. This section would present a vertical structure that towers over the rest of the draft. In terms of the sequence of events, if the author worked more linearly through the text, first polishing the beginning of the text and then progressively moving on to subsequent sections, the cloud would grow along a diagonal line from bottom left to top right. This shows that while the experience of a text is linear for a reader, this is not how writers create texts.

While edits are not especially localized, they are also not distributed completely evenly throughout the text. Using the information on the position of each edit, we can count the number of revisions each sentence has undergone. What we find is that authors work more on the first part of the text, with a maximum number of edits around the $10^{th}$ sentence of their work (Fig 1, top left), about 30% of the way through the work. This result, although it should be considered in the context of the kind of manuscripts being developed in the workshops (8–10 paragraph long scientific prose) is a strong indication of an asymmetry between the beginning of the text and the end in terms of number of edits.

To quantify the heterogeneity of this behavior among the authors, we count the number of edits of each sentence, and, for each author, we compute the Shannon-Wiener index (SW) of the distribution of edits. A null value would mean that only one sentence has been edited over and over again. A version of the process where the sequence of edits is randomly shuffled generates a distribution we show in cyan (Fig 1, right panel, top). The distribution of the SW index across all participants is reported in pink, with histograms of each different edition of SEW to highlight the general overlap. These results highlight how the complexity of the editing process is intermediate between localized edits on a single sentence and the random case of uniformly distributed edits throughout the texts. Alongside this, the relatively broad distribution of SW indices points to a diversity of behaviors among different writers. In other words, while writers don't completely focus their efforts or distribute them completely evenly, there is also individual variation: some writers have more focused effort than others.

## Measuring exploration

During the writing process, exploration—or the initiation of a planning-translation sub-cycle during revision—is an elusive concept, especially from a quantitative point of view. Some ideas and rhetorical solutions explored by the author will not leave any trace on any version of the text, but many others do, even if they will not make it to the final version. To quantify the level of exploration, we measure whether the author has followed the most direct path from the first version to the last version. If the writer follows this "path of least resistance" very directly, this is an indicator that their original plan for the text went largely unchanged during the process of writing, or that they followed a single line of thought, and in general that they did not commit to major revisions. On the other hand, more meandering routes from the first version to the final draft indicate more exploration during the writing process—in other words, changes to the original plan during writing. For this analysis, we analyze changes at the character level when computing edit distance. In this case, we aim for a more fine-grained measure of the overall amount of effort devoted to exploration, rather than the linguistic level at which changes were made.

Knowing the final version of the draft, we can imagine many different processes that led to that version, each of which can be represented as a sequence of edits. For any version $t$ of a draft, we can define the edit distance to the first version ($d_{0t}$) and the last version ($d_{tf}$). Together with the distance between the first and the last versions ($d_{0f}$), a triangle will be formed. The set of points such that $d_{0f} = d_{0t} + d_{tf}$ is, by definition, the one where all the shortest sequences of edits lie that lead from the first version to the last. In other words, where a writer is moving more or less deterministically along the $d_{0f}$ line, they are fluidly translating a pre-specified writing plan. A writer moving entirely along this line would never enter planning-translation sub-cycles during the revision stage; however, since human writers do enter these recursive sub-cycles, we should expect $d_{0f} < d_{0t} + d_{tf}$ at some point during the writing process. The more extreme the difference between $d_{0f}$ and $d_{0t} + d_{tf}$, the more exploratory the writer has been.

## Results

We find (left panel, Fig 2) that authors reach the maximum distance from the shortest path around the middle of the writing process. Our findings suggest that, halfway through the writing process, 40% of the work of the average author will not be part of the final version. This result implies that authors explore a substantial amount of options during the writing process, for example, adding a lot of material that will disappear later.

We introduce an Exploration Coefficient, which encapsulates how much of an author's work directly contributed to the final piece. Let us define as $H(t) = (d_{0t} + d_{tf} - d_{0f})/d_{0f}$ the distance from one of the shortest sequences, normalized by the distance between first and last version (in order to compare different texts). In other words, this is how far off a writer was from the shortest path between two versions. Knowing the first and the final version, we can retrospectively measure $H(t)$ of each intermediate version $t$. Adding together the contribution from all versions, we get a measure of how much exploration happened. Since this measure would depend on the number of versions, we normalize by dividing it by the number of versions itself. Thus we define the Exploration Coefficient as $E = \frac{1}{t_f} \sum_t H(t)$, where $t_f$ is the number of versions; $E$ would be equivalent to zero along the shortest path. Of course, not all differences can be considered exploration in its most general meaning. For instance, because this measure considers edits at the character level, minor edits could be transcription events: spelling errors, corrections, or minor lexical changes. The right panel, Fig 2c reports the histograms of the Exploration Coefficients for all authors of all workshops and disaggregates over the three workshops. Among all authors, we observe an average Exploration Coefficient of 28%,

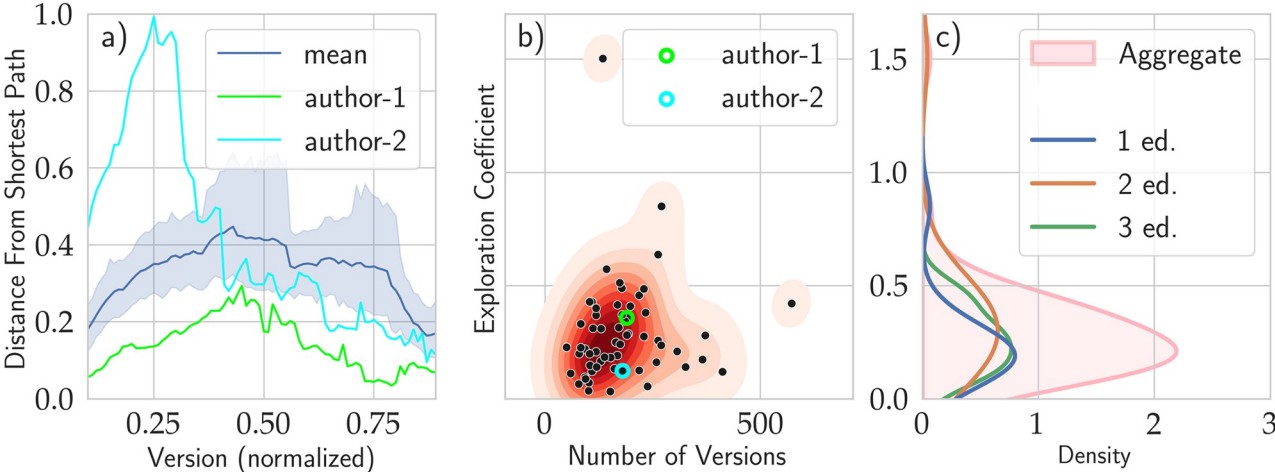

**Fig 2. Exploration patterns in writing.** (a) Distance from the shortest path during the editing phase from beginning to end for two authors [in cyan and green] and the average behavior of all the authors [dark blue line]. The shortest path would be achieved by someone who never makes revisions to the text. It is evident how the deviation from the shortest path varies strongly throughout the writing process, and it is stronger further from the beginning and the end. The mean curve is displayed as the dark blue line, while the pale blue area outlines the 99.5% confidence interval estimated using the bootstrap technique. (b) Exploration coefficient vs. the number of versions of each document. The Exploration Coefficient shows weak to no correlation with the number of versions, Pearson R = 0.10, p = 0.44. The values for the two versions presented on the left are outlined with circles of the same color. (c) Distribution of the Exploration Coefficient, defined as the area under the curves of the left panel, aggregated over all workshops [pink] and separately for each workshop.

meaning that, on average, an author changes 28% of what they have already written before the end of their writing process.

In principle, we expect authors who write more versions to have a higher Exploration Coefficient. The central panel, Fig 2b, allows us to check this hypothesis by comparing the Exploration Coefficient to the number of versions for each author. We observe that the correlation is slightly positive, though not significant ($\rho = 0.10$, p-value = 0.44). This means that the number of distinct versions is not an especially good predictor of whether an author is highly exploratory; authors with many hundreds of versions might explore only a fraction of the available space.

## Exploit or explore? The Twist ratio

The previous measure looked at exploration, an indicator of when a writer returns to the planning phase during revision. Here, we aim to measure a) when writers instead return to the translation stage, exploiting ideas and readily putting words onto the page, and b) when writers move between exploration and translation states. Below, we introduce a measure that indicates periods of effortlessness and absorption in the translation and transcription phase on the one hand, or periods of "twisting" between exploration and translation on the other.

Psychological flow is marked by absorption in a task, a subjective experience of time passing quickly, and a feeling of effortlessness [36]. As we write, we might find specific parts of the text coming easily—indicating a flow state during which we can exploit ideas, and satisfactory translation and transcription come effortlessly. Other parts of the text may need to be rewritten over and over again, indicating interruptions, dissatisfaction with how we have translated our ideas or dissatisfaction with the ideas themselves. Flow is largely defined introspectively; in writing research, by having writers report on their own experiences. As in other areas, flow in writing can be multidimensional and difficult to measure [37]. While periods of effortlessness

and absorption may be a positive indicator that a writer is subjectively experiencing the broader phenomenon of psychological flow, flow may take other forms during writing, and this would require further study. We thus refer specifically to "translation flow" as a state of effortlessness and absorption particularly in the sentence production phase.

To quantify this, we look at triplets of successive versions, for example, versions *A*, *B*, and *C*. Let us assume the ED distance between *A* and *C* is equal to the sum of the distances between *A* and *B* and *B* and *C* ($\overline{AC} = \overline{AB} + \overline{BC}$). In this case, when writing *C*, the writer hasn't over-written the changes they made when writing *B*, indicating the writer is in a translation flow state. Alternatively, if in *C* they have overwritten changes made in *B*, they are not satisfied with their work. In this situation, we say that the author is shifting back into planning and exploration, unable to maintain momentum in translation and transcription. If a writer is generating satisfactory text fluidly and without interruption, the trajectory in this space would be a straight line from the first version to the last. Deviations from this line imply that the author has made some changes with respect to the previous version, but that these changes may be unsatisfactory and short-lived.

With the Twist Ratio, we measure the relative number of inflection points between exploration and translation flow as angles in a Euclidean space between subsequent versions of the text. This procedure is possible since edit distance is a well-defined metric. As such, there exists an *N* dimensional space where each version can be represented as a point and all distances between versions are respected. We must, however, determine the number *N* of dimensions to use. High dimensional spaces will guarantee that all distances between versions can be preserved, but much of the structure is lost. For example we could chose $N = (\#\text{versions})^2$, in which case all distances are independent. Using principal component analysis (PCA), we established that over 90% of the information is captured in three dimensions.

Using the t-distributed Stochastic Neighbor Embedding, t-SNE, we can define a three-dimensional embedding where distances are preserved, with greater importance given to preserving short distances and thus the geometry of close points. As a result, we can define a complete trajectory followed by each text in Euclidean space on its way to its final form. An example of the trajectory of a text in the Euclidean space (and of a single triplet), is depicted in Fig 3. We show the trajectory in just two dimensions since most geometric properties appreciable by the eye are already present. Arrows indicate each step from one version to the next. The absolute values on the axes represent the position of each draft version in the embedding space, and as such, their specific values are irrelevant. Differences between two points show the character ED between two versions, i.e., the number of characters that must be edited for one to become the other.

In Euclidean space, the three distances form an angle, *β*, in *B*, shown in the inset in Fig 3. In the case of translation flow, the three versions will sit on a straight line and *β* = 180˚, while in the case of exploration −180˚ < *β* < 180˚. To better quantify the difference between exploration and translation flow, we arbitrarily define as exploration all events where the angle *β* is within the intervals [30˚, 150˚] and its opposite [−30˚, −150˚]. Correspondingly, we define translation flow events as those for which the angle is in the intervals [−30˚, 30˚] and [−150˚, 150˚]. This is shown in Fig 3 (inset), which shows the fixed locations of *A* and *C* (solid circles) and hypothetical versions of *B*. If the writer is in the area shaded in green, version *B* is sufficiently close to versions *A* and *C* that we can consider the writer to be in a translation flow state, exploiting existing plans. However, if *B* is far from both *A* and *C*—the area in blue—this is considered exploration. The Twist Ratio is thus defined as the fraction of versions in which the writer is in a translation flow state.

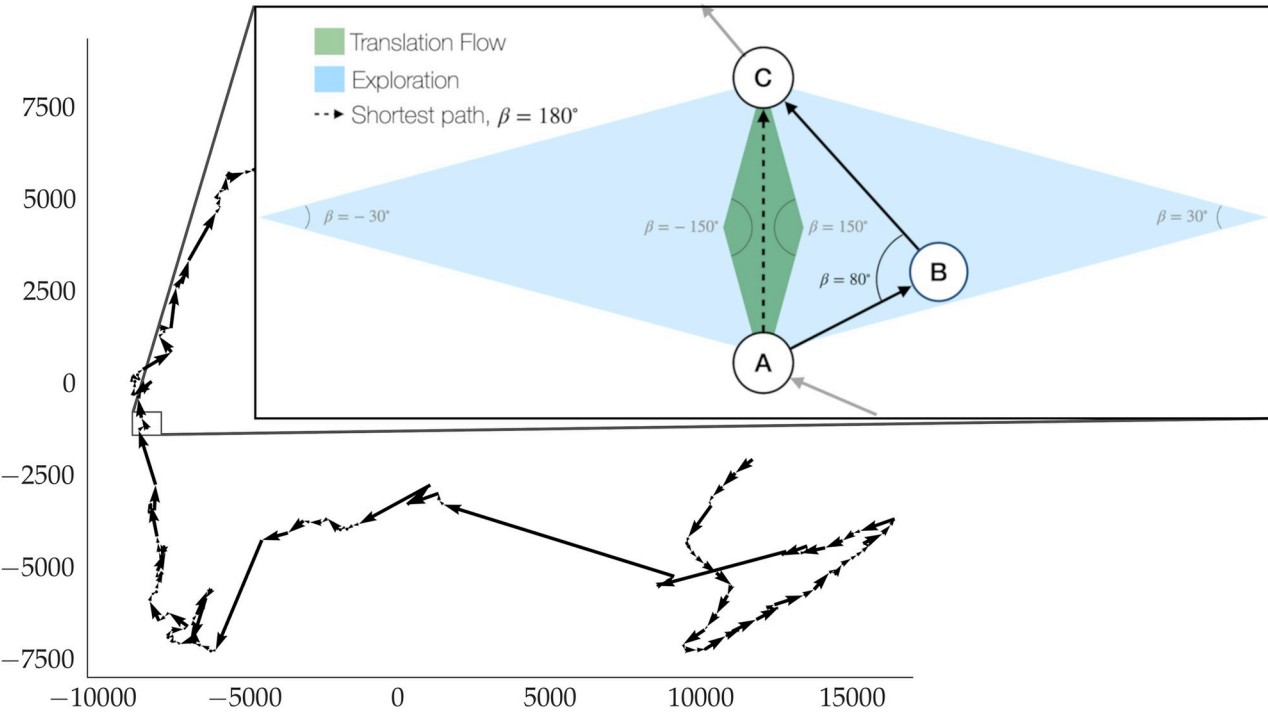

**Fig 3. Trajectory of a draft.** An illustration of the tSNE embedding of the draft versions for a single author in two dimensions. Distances in this space are the number of characters edited between one draft version and another. The set of arrows depicts the trajectory of the work on a single draft. The top right depiction shows an example section of the trajectory to display the vectors used in computing the Twist Ratio and the thresholds between exploration and translation flow.

## Results

In the left part of Fig 4, we report the distribution of the angles observed in all trajectories of all authors. We find that authors explore on average between 1% and 2% of the time, indicated by the area shaded in gray in Fig 4a. The dark blue line in Fig 4a indicates the distribution of average $\beta$ angles across authors (displayed as the distance between the shortest path, 180˚, and $\beta$), with most authors exploring around 1.6% of the time. In other words, writers spend most of their time in translation flow. Fig 4a shows the values for two specific authors, showing that some authors never enter a fully exploratory state: for example, author-1 only ever veers about 10–20˚ off the "path of least resistance" between draft versions. As with the other measures, this shows considerable individual variation; however, Fig 4b shows that this is more or less consistent across workshops.

Finally, the Twist Ratio decreases systematically with the total number of edits undertaken to reach the final product ($\rho = -0.46$ and p- value $< 0.001$). In other words, writers who edit less seem to explore more. Fig 4c shows that higher twist ratio values are more likely to occur for texts with fewer overall edits. This figure also shows the writers from Fig 4 an outlined; the exploratory forays undertaken by author-1 occurred with fewer overall editing events than author-2. In other words, author-2 may have done more editing, but author-1 explored more.

To conclude this section on the results of our investigations into the defined metrics we report the spearman correlation and significance among the three in Table 2 (for further information on the relation among metrics and other draft characteristics see Fig 1 in S1 File).

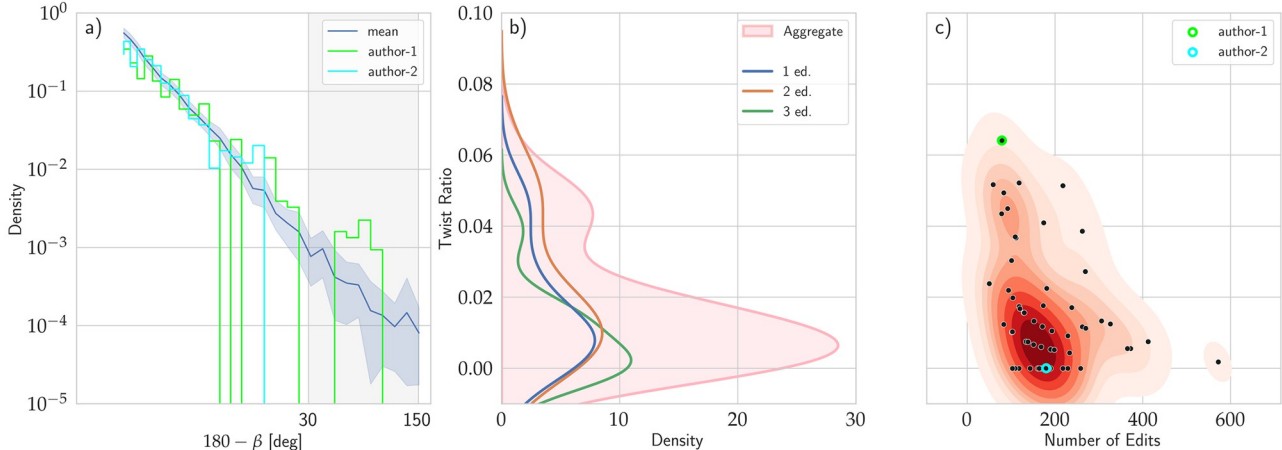

**Fig 4. Twist ratio.** (a) The distribution of the angle between consecutive velocities in the evolution of the draft. The area we have defined as exploratory is highlighted in gray. The dark blue line depicts the average distribution of the angle, while the pale blue area outlines the 99.5% confidence interval estimated using the bootstrap technique at each point. The light green and cyan lines show two example versions. (b) The distribution of the Twist Ratio over all workshops [pink] and the kernel density estimation for each workshop. (c) The distribution of the exploratory steps per draft compared to the number of edits for each workshop. The examples presented in the left panel are outlined in the corresponding color.

## Discussion & conclusions

We have proposed here a new set of methods for analyzing the writing process that relies on common versioning systems. These methods can give us detailed information about the author's activity, such as where and when they worked, and how much they have explored new pathways vs following a pre-planned route. Overall, this gives us a sense of how random, mechanistic, or complex the whole process is. The new metrics we introduced give a quantitative account of all of these aspects, providing key insights into the writing and revision process using simple edit distance measures on snapshots of writing activity.

The complexity measurement provides a way of seeing the spatial distribution of the writer's effort across the text. The observed distributions situate effort in our particular writing task between two extremes. Writers tended not to work intensively on one part of the text (which would give a complexity score of zero), but their effort also wasn't distributed randomly across the text. Normalized complexity scores were almost at the midpoint between these two extremes, indicating bursts of effort across the entire text, with a low peak about 30% of the way through. Moreover, the distribution of complexity scores across authors is heterogeneous—while no writers focus their efforts or distribute them completely, some are more focused than others, while others show more distributed effort.

Our data also show that although texts are sequential entities for a reader, writers don't produce them this way. Not only did different parts of the text receive different levels of effort, but this effort wasn't sequential—writers aren't working their way from the introduction through to the conclusion. Future research could use our measures to explore what factors might mediate this individual variation, and how this variation might correlate to the properties of the finished text. For example, it may be that more skilled writers distribute their efforts more evenly across a text, while less experienced writers become more "hung up" on introductory paragraphs at the outset. The measures we introduce here could also be combined with more qualitative approaches like protocol analysis to discover whether the experience of phenomena like writer's block corresponds to the distribution of effort across a text: do we spend more or less

effort on parts of a text that we feel we are struggling with? Do we struggle more when we try to work sequentially on a process that is more naturally distributed?

Turning to the process of revision, we introduced measures that are potential indicators of shifts into and out of planning or exploration on the one hand and translation and transcription on the other. The distribution of the Exploration Coefficient confirms that writing is not a strictly linear process from planning to sentence generation to revision; effective revision entails recursive cycles of exploration and translation. If the process were linear, we would observe lower values of the coefficients, indicating an author who sticks to a preconceived plan and deviating minimally from the shortest path between draft versions. Instead, we observe the opposite. The mean distance from the shortest path reaches its highest value (40%) around the middle of the process and doesn't decrease significantly until the end of the work (85 − 90% of the version history). This result implies that, until the final revision events, almost half of the text has to change to reach the final product. The observed behavior is not compatible with a model where planning the text and getting the words out it are two discrete activities. While earlier work has acknowledged this, measuring it has proved more elusive; our exploration coefficient provides a way to detect and quantify adaptations to the overall plan of a text in the process of revision.

Individual exploration profiles are a very diverse set of curves with a high degree of volatility. The volatility indicates that restructuring events often occur in the writing process. However, the timing of these events can offer different explanations for their underlying cognitive activities. For example, revision at the beginning of the activity might indicate difficulties with getting started, while revision at the end could serve a different purpose, such as improving coherence. It is noteworthy that for the distribution of turning points in an author's work, we find a heavy-tailed distribution, with more radical changes being increasingly rare. The fact that the Twist Ratio decreases with the number of edits suggests that exploration becomes more difficult as more work is put into the draft, probably because of a higher number of constraints. The complexity measures showed that effort is not invested in the same sequence that a text is experienced by a reader, and that effort was not evenly distributed across the text. This is echoed in the temporal dynamics of how a writer approaches their final version: they are not necessarily getting equally closer to their final version with each edit, but approach the final version more rapidly as the process concludes. The three viewpoints of the metrics discussed are substantially complementary as demonstrated by the correlations shown in Table 2, with the only significative correlation being small and negative among the exploration coefficient and the twist ratio.

While our results show an interesting new way of looking at the phases of writing, there are, of course, limitations. First, the genre may have heavily influenced the patterns we observed. Short scientific texts require accurate framing to best support the core conclusions that sit in the main body of the text, which could explain our writers' effort being focused more on the beginning of the text. This effect could also be ascribed to writers' perceptions of the reader's attention, assuming more precision is needed at the beginning of a text to ignite a reader's interest. Length of both the workshop and the broad limits of the text length may also have affected the results. It remains to be seen the extent to which these dynamics operate on more

**Table 2. The spearman correlation among the metrics discussed in previous sections and their significance.**

| Metric 1 | Metric 2 | Correlation | P-value |
|---|---|---|---|
| Complexity | Exploration Coefficient | −0.14 | 0.3 |
| Complexity | Twist Ratio | −0.002 | 0.99 |
| Exploration Coefficient | Twist Ratio | −0.33 | 0.01 |

prolonged writing processes and texts; for example, the broad peak of edits around the 10th sentence, about 30% of the way through the texts in our sample, may be an artefact of the length. This could reflect a focus on the first third of a text, or it could reflect a narrower focus on introductory paragraphs. We might expect other genres of writing, like poetry or song writing, to have completely different dynamics. Nonetheless, these measures provide us with new ways of testing predictions about how revision works across genres, levels of experience, and scales of text.

Finally, the methods we discussed are also applicable to collaborative works that could present very different dynamics to those observed in single authors. Indeed, scientific publications in particular are often the work of multiple collaborating writers. The measures we introduce here could be applied to a single text with multiple authors, revealing where different authors concentrate their efforts within the same text. A complex challenge in co-authoring as a writer is understanding when and where to concentrate effort given that others are simultaneously working on the same text. Understanding the dynamics of a Author A's twist ratio, for example, could signal to Author B that a particular part of a complex text is nearing completion, and ready for review.

In years to come, the writing practices are going to be deeply influenced by the new AI-powered NLP technologies. Current writing assistants offer corrections and alternatives, but do little to improve the author's own writing process. The metrics we introduce give an innovative account of this process, which could be applied in writing tools. These tools would be capable of suggesting to the writer which part of the text has been neglected, highlight when the writer is in the midst of effective translation flow, or when the writer is "twisting" and might do well to take a break. In other words, these tools, designed around a deeper comprehension of the human cognitive process of writing, could help writers to fully reach their potential as authors.

## Supporting information

**S1 File.**
(PDF)

## Acknowledgments

The authors wish to warmly thank Vittoria Colizza, with whom the SEW project was conceived and bootstrapped. The authors also wish to warmly thank Mark Buchanan and Justin Mullins for their invaluable contribution during the SEW workshops and a neverending series of insightful conversations. Last but not least, warmful thanks go to Elisabetta Falivene (Sapienza University of Rome) and Riccardo Corradi (iLab) for their invaluable technical and organizational contributions in running the SEW workshops. The following institutions provided support for the organization of the SEW workshops: Sony CSL Paris, Institut National de la Santé et de la Recherche Médicale (National Institute of Health and Medical Research of France), Sapienza University of Rome, Complexity Science Hub Vienna (Prof. Stefan Thurner), The Albert Einstein Center for Fundamental Physics in Bern (Prof. Gilberto Colangelo), the Institut des Systèmes Complexes in Paris (Prof. David Chavalarias).

## Author Contributions

**Conceptualization:** Donald Ruggiero Lo Sardo, Pietro Gravino, Vittorio Loreto.

**Data curation:** Donald Ruggiero Lo Sardo.

**Formal analysis:** Donald Ruggiero Lo Sardo, Pietro Gravino, Christine Cuskley.

**Funding acquisition:** Vittorio Loreto.

**Investigation:** Donald Ruggiero Lo Sardo, Pietro Gravino.

**Methodology:** Donald Ruggiero Lo Sardo, Pietro Gravino, Vittorio Loreto.

**Project administration:** Pietro Gravino, Vittorio Loreto.

**Resources:** Vittorio Loreto.

**Software:** Donald Ruggiero Lo Sardo.

**Supervision:** Pietro Gravino, Vittorio Loreto.

**Validation:** Donald Ruggiero Lo Sardo, Pietro Gravino, Christine Cuskley.

**Visualization:** Donald Ruggiero Lo Sardo, Christine Cuskley.

**Writing – original draft:** Donald Ruggiero Lo Sardo.

**Writing – review & editing:** Donald Ruggiero Lo Sardo, Pietro Gravino, Christine Cuskley, Vittorio Loreto.

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
