## [Decision Letter · Decision Letter 0]

19 Jul 2022

PONE-D-22-13798Evolutionary-like dynamics of writingPLOS ONE

Dear Dr. Lo Sardo,

Thank you for submitting your manuscript to PLOS ONE. After careful consideration, we feel that it has merit but does not fully meet PLOS ONE’s publication criteria as it currently stands. Therefore, we invite you to submit a revised version of the manuscript that addresses the points raised during the review process.

The manuscript has been revised by two experts in the field. Both  Reviewers appreciated your work in terms of the potential contribution at the empirical and theoretical level. However, they both  found some critical issues that prevented me to consider this version of the manuscript ready for publication. I agree with them that the measures you introduced need to be better described and discussed, also with respect to other measures present in the literature; in addition, the study needs to be more appropriately situated at the theoretical level. Finally, but very importantly, I recommend to make data on which analyses are based available, clarify what type of consent has been asked to participants and whether the study has been approved by an ethical Committee .

At this stage, I am inviting you to submit a revised version of the paper for consideration.  In doing so, please carefully address each of the reviewers' comments and outline the changes you have made in cover letter to me.

We look forward to receiving your revised manuscript.

Kind regards,

Francesca Peressotti, Ph.D

Academic Editor

PLOS ONE

Journal Requirements:

“The following institutions provided financial and organizational support for the organization of the SEW workshops: Sony CSL Paris, Inserm, Sapienza University of Rome, Complexity Science Hub Vienna (Prof. Stefan Thurner), The Albert Einstein Center for Fundamental Physics in Bern (Prof. Gilberto Colangelo), the Institut des Syst`emes Complexes in Paris (Prof. David Chavalarias).”

“SONY CSL Paris funded the work of Donald Ruggiero Lo Sardo, Pietro Gravino and Vittorio Loreto

La Sapienza university of Rome funded the work of Donald Ruggiero Lo Sardo and Vittorio Loreto

“No”

Reviewers' comments:

Reviewer's Responses to Questions

**Comments to the Author**

1. Is the manuscript technically sound, and do the data support the conclusions?

Reviewer #1: Partly

Reviewer #2: Yes

2. Has the statistical analysis been performed appropriately and rigorously? 

Reviewer #1: N/A

Reviewer #2: N/A

3. Have the authors made all data underlying the findings in their manuscript fully available?

Reviewer #1: No

Reviewer #2: No

4. Is the manuscript presented in an intelligible fashion and written in standard English?

Reviewer #1: Yes

Reviewer #2: Yes

5. Review Comments to the Author

Reviewer #1: My name is Mark Torrance, I think authors should know who is reviewing their work. My expertise is in the educational and cognitive psychology of text production. I have quite extensive experience of interpreting real-time data from text composition but no direct experience of developing the kinds of revision-analysis methods reported in this paper.

The authors of this paper provide three metrics for capturing how a text grows and develops as a writer or writer moves from first word through production and editing, to a final text. These measures draw on versioning methods that compare editing distance and so are semantically, linguistically, and chronometrically agnostic. They illustrate the potential of these metrics in an analysis of a relatively large sample of writing process data from academic writers producing demanding texts.

I am pleased to see a paper with this focus submitted to a multidisciplinary, general interest journal. I see value in the metrics that they report - I think they make a useful and potentially important contribution - although I would need to invest more time than I currently have to work though in detail what they provide. I also agree with the authors that this general approach - exploring in meaningful ways variation across texts in the extent to which they are manipulated / modified during their production - helps develop out understanding of the mental (and possibly social) processing that underlies human text generation.

I think, therefore, that it should be considered for publication. My concern, however, is that the authors currently make unnecessarily inflated claims about the scope of their work. It is not necessary (or correct) to represent it as offering a radical new research paradigm for studying writing. It is also not necessary to couch their work in buzz words like "evolutionary". What they are doing is (a) offering what are (I think) robust methods for capturing one aspect of writing process that, combined with other approaches, will help to build cognitive theory around text production, and (b) introducing the possibility of rigorous science of text production to a new audience. These are both valuable goals. But their paper will only achieve them if (a) they adequately situate their work in existing science and (b) the provide readers with a stronger feel for the constructs that they are measuring. So less hyperbole and more basic science.

I therefore want to make just three broad points. I'm hoping that the authors will be give the opportunity to revise their paper in light of these, and will be very happy to comment more specifically on the detail of their proposed metrics. (Please excuse typos - I'm rushing to meet the Journal's tight deadline.)

1. Theorising in the introduction is imprecise and insufficiently rooted in existing theory and research. There is too much grand gesture here, I think, and too little incremental science. I hope the following two examples illustrate what I mean. "There is ample evidence that the writing process influences the quality of the final text and that mastering self-regulating strategies leads to higher coherence and, in general, better texts [3–12]. " The first part of this claim is trivially true. Text only results from the processes used to create it. There is not other way in which it appears on the page. The authors might here be thinking just about the particular strategic activity that a writer engages in - the ways in which they choose to regulate their own behaviour as they produce text affects text quality independently of their content knowledge and linguistic skills. If so, that's what they need to say. However, that's an empirical claim, and there's counter evidence. The list of papers referenced here contains a whole range of research, including papers on the effects of first grade instruction - a literature which does not come down strongly on the side of students needing explicit writing strategies (I would and have argued). It also includes one of my papers where our main finding is actually no relationship between process and text quality.

"As such, many have argued that writing is a complex task. The writer has a mental model of her/his work that must adapt since the whole is affected by variations of the constituent parts, such as individual sentences or even words [13]. We postulate that these adaptations trigger shifts from one writing phase to another. During these shifts in the cognitive task, the model of the text can undergo creative reconstruction. The identification and analysis of these events, as well as the study of other patterns of the process, can be crucial to improving writing strategies." Several claims here that need to be properly worked through and situations in existing research and theory. (1) "The writer has a mental model of their work." I am not sure what is meant by "work" here. This could be one or both a representation of the macrostructure (or surface form?) of the text produced so far, or a message-level representation of what they aim to communicate, or what what they believe that they have communicated so far. There are other possibilities. It's also not clear that these representations actually exist or play an important role in written production. They may do, but the evidence isn't there, to my knowledge. See, for example, https://doi.org/10.1017/S0140525X04000056 - and I have argued that similar processes are likely to occur in writing. It is possible that the macrostructure that makes text cohere into a message is emergent rather than something that maps directly onto any mental model, even one that has changed during writing. (2) "that must adapt since the whole is affected by variations of the whole is affected by individual sentences…or even words" Again, I'm not sure what this means. Words and sentences do not just appear. They are output from the writer's mental representation (at whatever level, assuming this exists) and so should necessarily cohere with existing representations. If the writer was a reader then this claim would make perfect sense. There may be a sense in which it is true for writers, but this is a theoretical claim. If it's important, then it needs unpacking and supporting. If it's not then if should be dropped. (3) " We postulate that these adaptations trigger shifts from one writing phase to another. During these shifts in the cognitive task, the model of the text can undergo creative reconstruction." I'm not sure what the authors are thinking of here. Something like you output a particular sentence that has a meaning that surprises you (for some reason). This then means that you decide to revise your text in light of new ideas? Are there other possibilities? If not, then just say this. If there are, then elaborate. Then, why does the shift in the task result in reconstruction (I'm assuming of mental representation of message? Isn't it the other way round - you change your ideas and so see the need to revise your text. And so on. My point here is definitely not that the authors should present a full blown theory of mental processing underlying text production, or that this is necessary, or that the authors intended meaning is wrong. They do, however, need to avoid vague hand-waving. If they are going to make a priori claims about mental processing underlying text production these need to be explained clearly and precisely - avoiding imprecise use of jargon - rooted in existing theory and research, and as far as possible evidenced.

2. The authors should carefully situate the new methods that they described in previous efforts to extract similar information from writers text-generation processes. "Here for the first time" in the following - "Here, for the first time, we uncover those processes by recording, through a suitably dedicated software, all edits and actions in the making of a text, from the early attempts to the finishing touches." - is not justified, I think (or at minimum the authors need to strongly evidence this claim). To be clear, I think the methods that they describe are of potential value, and I would be excited to see them reported in a general-interest journal. However, they do not represent that paradigm shift that the authors are claiming. In particular, they are not an alternative to approaches involving think aloud (although it's a long time since I've seen papers using these methods), experimental manipulations, and keystroke logging. These are currently represented as alternative approaches to a single research goal. In fact, they serve quite different purposes, and are not in any sense mutually exclusive. This is not an appropriate framing for their new methods. What the authors want to demonstrate is how texts evolve over time, particularly in contexts (relatively rare but important) where the text is subject to extensive revision. This is an absolutely worthwhile goal. It is, though, something that has been addressed before. I've not used these methods in my own research, but if I was going to I would search forward from Eklundh, K. S., & Kollberg, P. (2003). Emerging discourse structure: computer-assisted episode analysis as a window to global revision in university students’ writing. Journal of Pragmatics, 35(6), 869–891), which is implemented in Inputlog (cited in the current paper), Also graph-based methods (see https://doi.org/10.1075/z.194.09leb;
https://www.ggxlog.net/). Another example is the choice of revision unit problem that they authors address from lines 133 onwards. Their solution sounds good, but the problem has been addressed in previous research (I think) - see papers here https://research.tue.nl/en/persons/ma-rianne-conijn/publications/ . I think a review of this literature needs to be the main focus of the authors' introduction. I am definitely not claiming that the authors' cover the same ground as these, but their focus needs to be on how the methods the they describe move beyond these.

3. The measures that the authors report are interesting and potentially valuable. On the basis of what they report I do not get a strong feel for construct validity. This is for two reasons. First, the authors are vague about the construct that they aim to measure. It feels at times that the measure is driving the construct and not the other way round. For example I am not clear what it means for a writing process to be "structured" as in "The Complexity Score aims to quantify how structured the writing process is." (line 134). The authors illustrate this with "only one sentence was edited repeatedly, the process is highly organized. If all sentences were edited a comparable number of times, the process is indistinguishable from a random one" (line 162). I can see how these two cases are different, and that much more revision occurs in the latter case, but I'm not sure why the second one is less structured or organised. The second reason is that, even as a reader who is used to abstract representations of texts and writing processes, I find it very difficult to relate indices and plots to my understanding of how texts might change or how writers might engage in cycles of revision. I don't feel that I have a strong enough understanding of the measures that the authors propose to make recommendation for how this gap might be bridged, but something is needed. Again, to be clear, I am not at all criticizing the authors for seeking to abstract features of the ways that a writer or writers interact with and develops their texts and specific indices. I just need a stronger feel for what is being measured.

Reviewer #2: Summary

The authors propose a new set of measures to describe text writing processes in the manuscript. The measures are: a) the Complexity Score, which accounts for the complexity encountered by the writer in building the text, measured as the distribution of changes along text versions; b) the Exploration Coefficient, that is the degree of exploration of ideas, measured as the edit distance of a version from the initial and the final versions (the shortest path); c) the Twist Ratio, as the turning points in the flow of thoughts of the writer (exposition and exploration pattern).

These measures were explored in a sample of texts produced by young researchers who attended a workshop on scientific writing. Conclusions stress how these measures can be useful for modeling the writing process.

I found the paper's topic interesting and the research itself appears to be conducted carefully. I applaud the authors for attempting to develop metrics able to quantify such complex behaviors during text production.

I have some advice for improving the exposition which I have found good, but somehow difficult to follow mostly because complex metrics are described in the methodological section without a clear introduction on their meanings and aims. I report below my observations according to the manuscript sections.

Abstract

In the final sentence, I suggest substituting the word "learning" because the study does not treat learning in the writing process.

Introduction

The introduction is overall well-structured and easily readable by a large audience. It makes a good, even if synthetic, overview of the state-of-the-art methods for analyzing text writing processes, from natural language processing to cognitive sciences works.

My impression is that references are scarce. For example, in line 26 there is a list of fields that studied text analysis but only one reference. The following paragraph on cognitive sciences presents only one reference for each method, even if the scientific works are many. Here, I would use a different term instead of "pedagogy" (line 34). Studies are generally concentrated on learning progress during the education path or in analyzing the writing level in a specific grade. "Pedagogy" is more related to "educational facts", while most of the cited studies are in the field of developmental and educational psychology or general psychology and linguistics.

In the same paragraph (lines 47-48), it is not so clear to me why the authors compare keystroke logging to technics such as functional magnetic resonance, GRS, EEG. These technics are common to cognitive psychology but are quite different from each other. I would omit that sentence.

The last paragraphs of the introduction present the study and the outline of the following sections.

I would suggest focusing more clearly on the aims of the work (analyzing complexity, exploration, and exploitation in text generation) and briefly introducing the three original metrics developed.

It seems that the introduction lacks presenting the intents of the study or, in other words, why those metrics are chosen to analyze the writing process. A more detailed explanation of how each metric is devoted to analyzing a specific behavior would make it easier to read the methodological section. Only in Results the reader has a clearer idea.

Finally, in line 62 "we also present a new dataset composed of …". I suggest rephrasing given that the authors are not presenting a dataset. As I understand, the data will be not public for privacy reasons. This poses some doubts about the absence of an ethical statement and the collection of personal information.

Minor points: line 85 and line 88, substitute "section" with "subsection". Line 83, introduce ED as an acronym for Edit Distance.

Materials and Methods

Experimental Setup. Table 1 presents the number of participants in the workshop. However, the authors declare that "the analysis includes only data for participants who's (whose?) draft change at least 10 times". How many participants were discarded? Please, report the final sample size.

Measuring Complexity. Line 146, I suggest introducing the scales you tested. The reader understands only later that are characters, words, and sentences.

Measuring Exploration. Please, define "shortest sequences" and shortest path".

The last sentence of the subsection is not clear and poses doubts on the choice to use characters and not sentences as for the complexity measure. Please define "the structure of individual change".

Results

Overall, the introductory paragraphs of each subsection are discursive and useful to understand the aims of the metrics. It sounds unusual to read in the results section such useful information that would be precious also to understand the methodological section. I would suggest anticipating this information in the introduction of the study.

Exploration. Lines 282-284. Are the minor edits counted in the analysis of exploration? Could this be a limit of the metric?

Discussion

The discussion is well written and gives a good overview of the findings and the potentialities of the metrics.

I was wondering if the heterogeneity of the complexity score (discussed in line 364) would be due also to differences in L2 levels. Less proficient L2 writers probably necessitate more time in revising and restructuring sentences, for example.

Overall, the final discussion is easy-to-read and open to interesting possible applications of the metrics.

Conclusions

Line 445, "discuss university" seems an improper insertion.

6. PLOS authors have the option to publish the peer review history of their article (what does this mean?). If published, this will include your full peer review and any attached files.

Reviewer #1: **Yes: **Mark Torrance

Reviewer #2: **Yes: **Tania Cerni

---

## [Author Response · Author response to Decision Letter 0]

16 Aug 2022

Rebuttal letter for “How do authors navigate the complexity of the writing process?”

We are thankful to the Editor for providing constructive feedback, and for her interest in allowing this submission to continue. We are also grateful to the Reviewers for providing insightful and detailed feedback that has helped to significantly improve the clarity and depth of our study. Our replies are provided through indented text beneath each comment from the Editor and Reviewers. When our replies incorporate the direct revision from the manuscript in quotations, we further indent these quotations within our reply.

Once again, we are grateful for the opportunity to submit our revised manuscript to PLOS ONE, and we are happy to readily implement any additional improvements that arise in the review of our updated submission.

Response to the Editor

We are very thankful to the editor for considering our work and for the possibility to address the points raised during the reviewing process.

However, they both found some critical issues that prevented me to consider this version of the manuscript ready for publication. I agree with them that the measures you introduced need to be better described and discussed, also with respect to other measures present in the literature; in addition, the study needs to be more appropriately situated at the theoretical level. 

We thank the editor for their interest in our work. Regarding the contextualization of the metrics, their description and discussion we have edited the introduction adding more bibliographical references, and a paragraph aimed at better clarifying the scope of these measures. Following reviewer-2’s comment, we have moved part of the results, dedicated to the introduction of the metrics to the general public, to the methods section. We further address the reviewer’s comments in detail in the following sections of this document. 

Finally, but very importantly, I recommend to make data on which analyses are based available, clarify what type of consent has been asked to participants and whether the study has been approved by an ethical Committee.

We decided to follow your recommendation and to make the data on which our analyses have been performed available. They can be found here: https://github.com/SonyCSLParis/writingcomplexity.

During the registration, participants were informed about the data gathering and about the aims of the experiment and they accepted to:

- release the rights on the whole production at the workshop to Sony CSL, Sapienza and Inserm for scientific purposes only (they keep the intellectual property rights on my production);

- allow the use of my personal data for scientific purposes only, provided that they are not disseminated other than in aggregate form.

Finally, all the experiments were conducted in accordance with the organisers ethical guidelines (SONY, Sapienza University, and Institut National de la Santé et de la Recherche Médicale, i.e. INSERM of France). Furthermore, the procedures were validated by the SONY Computer Science Lab internal management board. The experiments were also reviewed and approved by each hosting institution: Complexity Science Hub Vienna (Prof. Stefan Thurner), The Albert Einstein Center for Fundamental Physics in Bern (Prof. Gilberto Colangelo), the Institut des Systemes Complexes in Paris (Prof. David Chavalarias). 

Response to Reviewers

We wish to thank the attentive work of the reviewers. We strongly believe that thanks to their suggestions and indications our work is much clearer and better situated in the wider conversation on the writing process. We took their advice and insight to heart and propose a series of changes that we describe in detail in the following.

Reviewer #1:

1. Theorising in the introduction is imprecise and insufficiently rooted in existing theory and research. There is too much grand gesture here, I think, and too little incremental science. I hope the following two examples illustrate what I mean. "There is ample evidence that the writing process influences the quality of the final text and that mastering self-regulating strategies leads to higher coherence and, in general, better texts [3–12]. " The first part of this claim is trivially true. Text only results from the processes used to create it. There is not other way in which it appears on the page. The authors might here be thinking just about the particular strategic activity that a writer engages in - the ways in which they choose to regulate their own behaviour as they produce text affects text quality independently of their content knowledge and linguistic skills. If so, that's what they need to say. However, that's an empirical claim, and there's counter evidence. The list of papers referenced here contains a whole range of research, including papers on the effects of first grade instruction - a literature which does not come down strongly on the side of students needing explicit writing strategies (I would and have argued). It also includes one of my papers where our main finding is actually no relationship between process and text quality.

We thank the insightful advice of the reviewer and have changed the text accordingly: 

Lines 12-15:

“The evidence on whether particular sequencing strategies, explicit or instinctive as they may be, contribute to the quality of the final text and whether mastering self-regulating strategies leads to higher coherence and, in general, better texts is the subject of ongoing research [3– 12]”

"As such, many have argued that writing is a complex task. The writer has a mental model of her/his work that must adapt since the whole is affected by variations of the constituent parts, such as individual sentences or even words [13]. We postulate that these adaptations trigger shifts from one writing phase to another. During these shifts in the cognitive task, the model of the text can undergo creative reconstruction. The identification and analysis of these events, as well as the study of other patterns of the process, can be crucial to improving writing strategies." Several claims here that need to be properly worked through and situations in existing research and theory. (1) "The writer has a mental model of their work." I am not sure what is meant by "work" here. This could be one or both a representation of the macrostructure (or surface form?) of the text produced so far, or a message-level representation of what they aim to communicate, or what what they believe that they have communicated so far. There are other possibilities. It's also not clear that these representations actually exist or play an important role in written production. They may do, but the evidence isn't there, to my knowledge. See, for example, https://doi.org/10.1017/ - and I have argued that similar processes are likely to occur in writing. It is possible that the macrostructure that makes text cohere into a message is emergent rather than something that maps directly onto any mental model, even one that has changed during writing. (2) "that must adapt since the whole is affected by variations of the whole is affected by individual sentences…or even words" Again, I'm not sure what this means. Words and sentences do not just appear. They are output from the writer's mental representation (at whatever level, assuming this exists) and so should necessarily cohere with existing representations. If the writer was a reader then this claim would make perfect sense. There may be a sense in which it is true for writers, but this is a theoretical claim. If it's important, then it needs unpacking and supporting. If it's not then if should be dropped. (3) " We postulate that these adaptations trigger shifts from one writing phase to another. During these shifts in the cognitive task, the model of the text can undergo creative reconstruction." I'm not sure what the authors are thinking of here. Something like you output a particular sentence that has a meaning that surprises you (for some reason). This then means that you decide to revise your text in light of new ideas? Are there other possibilities? If not, then just say this. If there are, then elaborate. Then, why does the shift in the task result in reconstruction (I'm assuming of mental representation of message? Isn't it the other way round - you change your ideas and so see the need to revise your text. And so on. My point here is definitely not that the authors should present a full blown theory of mental processing underlying text production, or that this is necessary, or that the authors intended meaning is wrong. They do, however, need to avoid vague hand-waving. If they are going to make a priori claims about mental processing underlying text production these need to be explained clearly and precisely - avoiding imprecise use of jargon - rooted in existing theory and research, and as far as possible evidenced.

We are thankful to the reviewer for indicating the lack of clarity in this passage of our work. We are hopeful that the revised version will clarify the claims of our work. 

Lines 17-26

“The writer has a rough idea, a model, of what her/his final output will be [13]. This model must adapt since the imagined final version must conform to variations of its constituent parts, such as individual sentences or even words [14, 15]. Examples might be rhetorical solutions that do not satisfy the author or instances where through the writing process some unforeseen clarity on the content or new idea is found by the author himself. We postulate that these adaptations trigger shifts from one writing phase to another, such as from sentence generation to revising or even planning. The identification and analysis of these events, as well as the study of other patterns of the process, can be crucial to improving writing strategies.”

2. The authors should carefully situate the new methods that they described in previous efforts to extract similar information from writers text-generation processes. "Here for the first time" in the following - "Here, for the first time, we uncover those processes by recording, through a suitably dedicated software, all edits and actions in the making of a text, from the early attempts to the finishing touches." - is not justified, I think (or at minimum the authors need to strongly evidence this claim). To be clear, I think the methods that they describe are of potential value, and I would be excited to see them reported in a general-interest journal. However, they do not represent that paradigm shift that the authors are claiming. In particular, they are not an alternative to approaches involving think aloud (although it's a long time since I've seen papers using these methods), experimental manipulations, and keystroke logging. These are currently represented as alternative approaches to a single research goal. In fact, they serve quite different purposes, and are not in any sense mutually exclusive. This is not an appropriate framing for their new methods. What the authors want to demonstrate is how texts evolve over time, particularly in contexts (relatively rare but important) where the text is subject to extensive revision. This is an absolutely worthwhile goal. It is, though, something that has been addressed before. I've not used these methods in my own research, but if I was going to I would search forward from Eklundh, K. S., & Kollberg, P. (2003). Emerging discourse structure: computer-assisted episode analysis as a window to global revision in university students’ writing. Journal of Pragmatics, 35(6), 869–891), which is implemented in Inputlog (cited in the current paper), Also graph-based methods (see https://doi.org/10.1075/z.194.;
https://www.ggxlog.net/). Another example is the choice of revision unit problem that they authors address from lines 133 onwards. Their solution sounds good, but the problem has been addressed in previous research (I think) - see papers here https://research.tue.nl/en/. I think a review of this literature needs to be the main focus of the authors' introduction. I am definitely not claiming that the authors' cover the same ground as these, but their focus needs to be on how the methods the they describe move beyond these.

We are thankful to the reviewer for helping us no, better situate our research in the context of revision and editing strategies. We believe our research provides an improvement upon some of the cited methods since it requires only versioning data and because in comparison to other graph techniques, the definition of a node is both flexible, and also grounded in the construct choice. We are hopeful that the changes made to the text will help better contextualise our work and also clarify elements of novelty in our research.

Author summary:

“Here we uncover those processes by recording, through a suitably dedicated software, all edits and actions in the making of a text, from the early attempts to the finishing touches.”

Lines 39-41, 62-64, 92-96.

“Existent work has used three main experimental setups that give access to different kinds of information on the writing process:[...]”

“Here, we propose a new method that presents a much smaller overhead than previous work such as S-notation [15], since it does not require keystroke logging to reconstruct the activity of the author.”

“In the subsection Exposing Complexity, we introduce a representation of the writing process that details where and when the author has made changes to her/his draft, and improves upon previous attempts by proposing a way to change scale from characters to words, sentences, paragraphs and other segmentations of text [28]”

3. The measures that the authors report are interesting and potentially valuable. On the basis of what they report I do not get a strong feel for construct validity. This is for two reasons. First, the authors are vague about the construct that they aim to measure. It feels at times that the measure is driving the construct and not the other way round. For example I am not clear what it means for a writing process to be "structured" as in "The Complexity Score aims to quantify how structured the writing process is." (line 134). The authors illustrate this with "only one sentence was edited repeatedly, the process is highly organized. If all sentences were edited a comparable number of times, the process is indistinguishable from a random one" (line 162). I can see how these two cases are different, and that much more revision occurs in the latter case, but I'm not sure why the second one is less structured or organised. The second reason is that, even as a reader who is used to abstract representations of texts and writing processes, I find it very difficult to relate indices and plots to my understanding of how texts might change or how writers might engage in cycles of revision. I don't feel that I have a strong enough understanding of the measures that the authors propose to make recommendation for how this gap might be bridged, but something is needed. Again, to be clear, I am not at all criticizing the authors for seeking to abstract features of the ways that a writer or writers interact with and develops their texts and specific indices. I just need a stronger feel for what is being measured.

We are thankful to the reviewer for highlighting the need for greater clarity both on the aims of our measures and their meaning. We hope the changes we made to the text will help the reader in these regards.

Lines 90-99, 190-199:

“The outline of the paper is as follows. In materials ad methods we discuss the data gathering procedures and the three original metrics introduced in this work and discussed in the results and discussion. In the subsection Exposing Complexity, we introduce a representation of the writing process that details where and when the author has made changes to her/his draft, and improves upon previous attempts by proposing a way to change scale from characters to words, sentences, paragraphs and other segmentations of text. We use this representation to investigate the distribution of work along the length of the draft and propose a measure of complexity that differentiates the space between randomness and ``machine-like'' writing, i.e., writing the final version directly without any revisions.”

“To clarify, when we talk of structure in the context of complexity, we talk of the structure of edits as mapped onto the unidimensional space of the text, i.e. the x-axis of Fig. 1 (b). If the author edits specific elements more than others, the editing process structure is simply organized around a few focal points. If the edits are more evenly distributed across the space of the text, they are the result of a more complex and less predictable process. It is important to remark that we are not looking at the dynamical process of writing, that happens in the space of the text and in time. We are focusing on the organization of the author’s attention across the text. The distribution of edits in the text space represents the distribution of the author’s effort. We use the effort as a proxy for the author’s attention.”

Reviewer #2: 

[...]

Abstract

In the final sentence, I suggest substituting the word "learning" because the study does not treat learning in the writing process.

We thank the insightful comment made by the reviewer and have updated the abstract accordingly:

“Our results and the new quantitative tools deployed have the potential to foster the discussion about the nature of writing and give the processes an evolutionary perspective, aiming at more creative and impactful writing processes.”

Introduction

[...] My impression is that references are scarce. For example, in line 26 there is a list of fields that studied text analysis but only one reference. The following paragraph on cognitive sciences presents only one reference for each method, even if the scientific works are many. Here, I would use a different term instead of "pedagogy" (line 34). Studies are generally concentrated on learning progress during the education path or in analyzing the writing level in a specific grade. "Pedagogy" is more related to "educational facts", while most of the cited studies are in the field of developmental and educational psychology or general psychology and linguistics. In the same paragraph (lines 47-48), it is not so clear to me why the authors compare keystroke logging to technics such as functional magnetic resonance, GRS, EEG. These technics are common to cognitive psychology but are quite different from each other. I would omit that sentence.

We are thankful to the reviewer for indicating the lack of clarity in these passages of the introduction. We hope the updated text will provide the needed clarification.

Lines 51-55

“Lastly, keystroke logging uses software that records each keystroke to characterize the evolution of the draft. Keystroke logging is an unobtrusive technique often used in conjunction with other observation techniques such as eye-tracking, functional magnetic resonance imaging (fMRI), galvanic skin response (GSR) and electroencephalography (EEG), to uncover relations between the writing process and the author.”

The last paragraphs of the introduction present the study and the outline of the following sections.

I would suggest focusing more clearly on the aims of the work (analyzing complexity, exploration, and exploitation in text generation) and briefly introducing the three original metrics developed.

It seems that the introduction lacks presenting the intents of the study or, in other words, why those metrics are chosen to analyze the writing process. A more detailed explanation of how each metric is devoted to analyzing a specific behavior would make it easier to read the methodological section. Only in Results the reader has a clearer idea.

We thank the reviewer for the suggestion and have edited the paper to better clarify the link between the chosen metrics, their explanation and the goal of each metric.

We have included a paragraph in the introduction to clarify the goals of the study and how the metrics contribute to that aim. The sections on each metric we have moved from the Results section to the Methods section in order to keep the structure of our work more streamlined. We are hopeful that with this change the reader is better equipped to understand the Methods section.

Finally, in line 62 "we also present a new dataset composed of …". I suggest rephrasing given that the authors are not presenting a dataset. As I understand, the data will be not public for privacy reasons. This poses some doubts about the absence of an ethical statement and the collection of personal information.

We are thankful to the reviewer for the comment. Regarding the dataset, we finally decided to provide anonymized information on the author in the form of edit sequences (as described in the method section), number of versions, the workshop label and the edit distance between all versions at this address:

https://github.com/SonyCSLParis/writingcomplexity

Regarding the consent, we have provided informed consent forms to each author for the scientific use of their data (see response to the editor).

Experimental Setup. Table 1 presents the number of participants in the workshop. However, the authors declare that "the analysis includes only data for participants who's (whose?) draft change at least 10 times". How many participants were discarded? Please, report the final sample size.

We are thankful to the reviewer for pointing out this lack of clarity. The numbers in the table refer to active participants, so participants that authored at least 10 versions of their text. We have updated the text to make this point clearer.

Lines 117-120:

“There were no exclusion criteria for the participants but the analysis includes only data for participants whose draft changed at least 10 times, to control for active participation (numbers in table 1 refer to active participants).”

Measuring Complexity. Line 146, I suggest introducing the scales you tested. The reader understands only later that are characters, words, and sentences.

We thank the reviewer for the suggestion. We are hopeful that the edited text states more clearly the scales we have tested.

Measuring Exploration. Please, define "shortest sequences" and shortest path".

We are thankful to the reviewer for pointing out the lack of clarity in this passage. We hope the changes to the text address these issues.

Lines 218-228:

“Knowing the final version of the draft, we can imagine many different processes that led to that version, each of which can be represented as a sequence of edits. [...] The set of points such that $d_{0f} = d_{0t}+d_{tf}$ is, by definition, the one where all the shortest sequences of edits lie that lead from the first version to the last. If, at a given time t, the author is not on one of the shortest sequences, then she/he has been, to some degree, exploring the conceptual space of her/his work. Let us define as $H(t) = (d_{0t}+d_{tf} - d_{0f}) / d_{0f}$ the distance from one of the shortest sequences, normalized by the distance between first and last version (in order to compare different texts)”

The last sentence of the subsection is not clear and poses doubts on the choice to use characters and not sentences as for the complexity measure. Please define "the structure of individual change".

We thank the reviewer for indicating the lack of clarity and hope that the updated text better explains our meaning.

Lines 232-235:

“For this analysis, contrarily to the previous one, we will treat each character separately when computing edit distance since we are aim to quantify the amount of effort that was devoted to this exploration, rather than the structural changes that where made to the draft.”

Exploration. Lines 282-284. Are the minor edits counted in the analysis of exploration? Could this be a limit of the metric?

We thank the reviewer for the interesting question. To clarify, minor edits are counted in the analysis of exploration. The size of the edits is the main reason their contribution is not seen as a major limitation. We envision a more precise metric that can distinguish between the two cases, that relies on edit classification that will be developed in future work.

---

## [Decision Letter · Decision Letter 1]

28 Sep 2022

PONE-D-22-13798R1How do authors navigate the complexity of the writing

process?PLOS ONE

Dear Dr. Lo Sardo,

Thank you for submitting your manuscript to PLOS ONE. After careful consideration, we feel that it has merit but does not fully meet PLOS ONE’s publication criteria as it currently stands. Therefore, we invite you to submit a revised version of the manuscript that addresses the points raised during the review process.

Dear Authors,

I received the comments of two Reviewers. Luckily, both Reviewers were the same who reviewed your paper in the first round. They found that you valuably revised the manuscript, and they clearly see some improvements. However, they both think that this version of the manuscript is not acceptable for publication. In particular, and I agree with them, the study must be situated in the theoretical framework. Pertinent literature should be reviewed, highlighting the innovative aspect of the method described, and its contribution should be explicitly outlined and discussed with respect to previous studies. The details of the methods must be clearly laid out and exemplified. I do not want to discourage resubmission but, since the main issues are almost the same raised in the previous round, I recommend the authors to resubmit their manuscript only if they could fully address them.

With many regards,

Francesca Peressotti

We look forward to receiving your revised manuscript.

Kind regards,

Francesca Peressotti, Ph.D

Academic Editor

PLOS ONE

Reviewers' comments:

Reviewer's Responses to Questions

**Comments to the Author**

1. If the authors have adequately addressed your comments raised in a previous round of review and you feel that this manuscript is now acceptable for publication, you may indicate that here to bypass the “Comments to the Author” section, enter your conflict of interest statement in the “Confidential to Editor” section, and submit your "Accept" recommendation.

Reviewer #1: (No Response)

Reviewer #2: (No Response)

2. Is the manuscript technically sound, and do the data support the conclusions?

Reviewer #1: (No Response)

Reviewer #2: (No Response)

3. Has the statistical analysis been performed appropriately and rigorously? 

Reviewer #1: I Don't Know

Reviewer #2: Yes

4. Have the authors made all data underlying the findings in their manuscript fully available?

Reviewer #1: Yes

Reviewer #2: (No Response)

5. Is the manuscript presented in an intelligible fashion and written in standard English?

Reviewer #1: (No Response)

Reviewer #2: Yes

6. Review Comments to the Author

Reviewer #1: My name is Mark Torrance and I was Reviewer 1 of the previous version of this paper. I appreciate the opportunity to review a revised version, and I remain convinced that there is value in the analytic methods that the authors present.

I need to be honest about my limitations, both in time and in competence. I am a writing researcher / cognitive psychologist / psycholinguist. I have used editing ed editing distance in my own research, understand versioning software reasonably well, and so forth. But I'm struggling. I think there is probably real value in the methods that are described but I don’t have the time to work through (or skill to quickly work through) them in detail and assess the authors' claims. I make three points below about areas that I feel need further attention. I recognise and appreciate that work that the authors have put into revising their paper, but in these areas I do not feel that they have gone far enough. However I don’t think my review really does justice to the meat of the paper.

1. The authors continue to make over-inflated claims for the method that they present. If this was the first paper to ever explore text composition processes, then this might possibly be appropriate. However that is not the case. The title and the first four sentences of the abstract illustrate this. An appropriate title would be "A new method for describing the changes that writers make to their texts during composition" (or a better version of this). The title that authors offer is "How do authors navigate the complexity of the writing process?" I am not sure what this really means. Something like "How do writers produce complex texts?" perhaps? But if so, that's not a question that could possibly be answered in a single study (or lifetime!). Note that this is not a criticism of (just) the title and the start of the abstract. It's a repeat of my request to recast the whole paper so that it is clear what the method that the authors presents offers to researchers interested in establishing patterns in how writers' texts develop. This requires carefully specifying the problem that needs solving, looking in some detail at existing methods, and then presenting the new method in such a way as to make clear what it is adding to what already exists. As I said above, I am convinced that their method does make a contribution, and my understanding is that PlosOne is very happy to present basic, incremental science (i.e. there is no need to inflate claims).

2. Unless the authors have strong supporting evidence then I think they should drop all claims about underlying cognitive processes. Their method is useful (I think) in describing a set of phenomena. It does not provide direct insight into the cognitive processes by which these occur. For example "This result is consistent with the idea that a sentence is the smallest complete unit of thought." (line 171). I don't know what a "complete unit of thought" might be - as a cognitive psychologist it's not a construct that I'm familiar with. There is the idea of "planning scope" in literature on spoken and written production, which might be the sort of thing that the authors are thinking of. However, there's a quite well developed literature here that suggests this is less than a clause, and certainly less than an orthographic sentence (string ending with a sentence-terminating punctuation) which I think is the unit that the authors are talking about here. Similarly here "We are focusing on the organization of the author’s attention across the text. The distribution of edits in the text space represents the distribution of the author’s effort. We use the effort as a proxy for the author’s attention." Both attention and effort have specific (and varied) meanings in cognitive theory. Exactly what is meant here is not clear. It either needs to be unpacked and evidenced or (better) dropped. Again, this is illustrative. I think the authors need to go through their paper and remove all unsupported assumptions about underlying process. The one exception to this is where the authors discuss findings from their own data. Here I think some conjecture about process is valuable, though it needs to be strongly caveated.

3. I am still struggling to understand the methods that the authors describe. This is partly because I feel that I need to spend a lot more time with them than I have available for this review. But partly because I think there are gaps in the authors explanations. For example - lines 152 to 172 - first paragraph of "measuring complexity". I understand how you can establish editing distance at the character level. However, if your unit of analysis is a sentence, then are two sentences that are different by one character and that are different by multiple words then treated as similarly different. The fundamental problem I'm struggling with is that whereas characters are themselves invariate (an "s" is always an "s"), words and sentences aren't. Authors don't just move whole sentences around. I'm clearly missing something here. But I don't think I can find it in the author's existing explanation. I need a worked example, or more detailed explanation to understand.

Reviewer #2: I wish to thank the authors for considering my suggestions. I would also reiterate my positive opinion about their work, especially considering the metrics they developed. Some aspects in the methodological section (such as participants and ethical issues) are solved.

However, in my opinion, they did not substantially improve the introduction of the manuscript as theoretical aspects are not well contextualized. As I stated in my previous comments, the introduction is not exhaustive, and the aim of the study and of the metrics is not easily deducible without going into the methodological section. The methodological section should be devoted to understanding the technical details of the metrics, not their theoretical functions which should be better described in the introduction. This is only partially done. I see in this new version a synthetic picture of the framework and of the study aims. Nevertheless, I suggest rethinking how to introduce the metrics keeping in mind the large audience that the manuscript could reach thanks to this journal.

In this vein what I recommend is to rethink the theoretical framework. As I previously wrote, I have the impression that references are scarce. Lack of references is not only (and not always) a matter of few citations, but it is also (and more importantly) a matter of anchoring the present study into a theoretical framework. For instance, text planning, editing and revising processes are previously studied. However, the literature review presented here does not say how the previous studies addressed and measured these processes.

A targeted example is: (line 22) "We postulate that these adaptations trigger shifts from one writing phase to another, such as from sentence generation to revising or even planning. The identification and analysis of these events, as well as the study of other patterns of the process, can be crucial to improving writing strategies". How did the cited literature in the next paragraphs (in natural language processes and in cognitive science) address (or did not) this "identification and analysis"? And why do the present metrics add new insights? Answering these questions would help in identifying why and in which framework the specific methods (the metrics) are developed, and so the aims and the merits of the work.

I take the liberty to report a sentence of reviewer #1 that probably better explain in a few words my impression: "It feels at times that the measure is driving the construct and not the other way round".

Some details I would like to underline:

Line 51-61. The authors re-formulate the sentence about keystroke logging and other techniques such as EEG, and fMRI. Now the reason why they mention these techniques is clearer. However, (1) such techniques are just mentioned without any citation of the studies they refer to. It also lacks to explain to us the contribution of these studies to text production analysis. Furthermore, the sentence: "Keystroke logging is not conceived to analyze text evolution as a whole or where the author focuses her/his effort." Is this really true? Can you justify the sentence? What do you mean by "text evolution as a whole"? I suppose that keystroke logging (also when powered with physiological or neural measures) is devoted to understanding deeply the cognitive effort of the writer. Knowing the studies that the authors are referring to would be helpful to disentangle these doubts.

Line 82-90. The added sentence helps in contextualizing the measures. I think that the authors should reformulate the expression "writing cloud" as they have not mentioned what is a writing cloud yet. Alternatively, they should define it.

To conclude, I sincerely thank the authors for their efforts in disentangling some passages of their manuscript. I encourage them to further review their work by reformulating the introduction keeping in mind the novelty of their contribution and the specificity of their field of study, but without forgiving to contextualize their work also in the framework of previous literature about theoretical and methodological aspects of text production.

7. PLOS authors have the option to publish the peer review history of their article (what does this mean?). If published, this will include your full peer review and any attached files.

Reviewer #1: **Yes: **Mark Torrance

Reviewer #2: **Yes: **Tania Cerni

---

## [Author Response · Author response to Decision Letter 1]

13 Jan 2023

Reviewer 1

We thank the reviewer for engaging with the work once again and providing valuable feedback. Below, we briefly summarise each of their numbered comments alongside how we addressed them.

“[the authors should] carefully specify the problem that needs solving, looking in some detail at existing methods, and then presenting the new method in such a way as to make clear what it is adding to what already exists”. 

We have done extensive rewrites to frame our methods within the well-accepted framework of writing as a process that takes place in three interrelated phases: planning, sentence generation (translation/transcription), and revision. Specifically, we have detailed how our measures are a new way of looking at cycles of planning and translation during the revision phase in the introduction, as well as referencing this model more concretely in explanations of our measures and results. While the interrelatedness of these phases is well accepted, we contend that they are often measured in isolation, and our measurements provide a new way to look at them in an integrated way.

“[the authors] should drop all claims about underlying cognitive processes…[including] a ‘complete unit of thought’ …[and] ‘effort as a proxy for the author’s attention’...[where authors discuss these issues with respect to their own data] it needs to be strongly caveated”

We have removed reference to units of thought and attention; we agree these are either unclear or not measurable in our dataset. 

We have, however, kept the notion of effort: we contend that particularly our complexity measure gives a good indication of where in the text a writer focused their effort, but have added an additional explanation that this is a spatial measure, not a temporal one (i.e., it doesn’t speak to how much time a writer spent making an edit, but where in the text they concentrated their editing efforts; lines 239-251). 

In terms of the sentence-level unit we use for the complexity measure, we’ve removed reference to “unit of thought”, but instead clarified that the sentence level is where, in our dataset, we can best capture related adjacent edits as a single event, and we point out that this is consistent with previous work indicating that the translation aspect of sentence generation integrates with working memory mainly at the sentence level (Kellogg, 2004)

“I am struggling to understand the methods that the authors describe…I need a worked example, or a more detailed explanation to understand”

We’ve added detail to the explanation of our methods throughout to add clarity, particularly in terms of the framing we introduce in the introduction.

We have restructured the manuscript so that the results for each measure are adjacent to the introduction of the measure itself. In addition to the detail added in (a), this makes the aims of each measure clearer with respect to our specific dataset.

We have updated Figure 3 in order to better explain the geometrical elements used to compute the twist ratio with a concrete example of an author’s draft trajectory

Reviewer 2

We thank the reviewer for their feedback, especially clarity about issues resolved in the first round of revision. We respond to their most salient points below

“The methodological section should be devoted to understanding the technical details of the metrics, not their theoretical functions which should be better described in the introduction. I suggest rethinking how to introduce the metrics keeping in mind the large audience that the manuscript could reach”

This comment was echoed by reviewer 1 (point 1); we have provided more detailed theoretical framing throughout the introduction and situated our results and metrics within this new framing.

“text planning, editing and revising processes are previously studied. However, the literature review presented here does not say how the previous studies addressed and measured these processes.”

We have now put previous literature in a dedicated section ‘Understanding writing’. We focus on the fact that while previous studies acknowledge the interrelatedness and non-linearity of the well-known three phases of writing (planning, sentence generation, revision), they often measure these in isolation. The motivation for our measures is therefore to look at these together - particularly at revision as sub-cycles of planning and translation.

“The authors re-formulate the sentence about keystroke logging and other techniques such as EEG, and fMRI…such techniques are just mentioned without any citation of the studies they refer to.”

We have now added specific supporting literature for this on lines 57-59.

“It also lacks [explanation of] the contribution of these [keystroke] studies to text production analysis…Keystroke logging is not conceived to analyze text evolution as a whole or where the author focuses her/his effort.’ Is this really true? Can you justify the sentence? What do you mean by ‘text evolution as a whole’? I suppose that keystroke logging (also when powered with physiological or neural measures) is devoted to understanding deeply the cognitive effort of the writer. Knowing the studies that the authors are referring to would be helpful to disentangle these doubts”

On lines 55-72, we clarify the focus of previous keystroke logging studies, which differs generally in its aims from our approach (which does not use keystroke logging in any event). We point out that it often isn’t used in isolation, but alongside other methods, and nonetheless is used to look at when writers focus their efforts in terms of the timecourse of their writing, and not where in the text they are focused (lines 69-72). We also return to this issue in the discussion, where we point out that like keystroke logging, our measures relying on versioning (which has lower overhead than keystroke logging) could be usefully combined with more traditional approaches like protocol analysis and task-driven experiments (lines 469-474).

“ I think that the authors should reformulate the expression ‘writing cloud’ as they have not mentioned what is a writing cloud yet. Alternatively, they should define it.”

We have made changes to the abstract to indicate a clear definition of what we mean by “writing cloud”, as well as flagging for the reader that this is a concept we introduce in the course of the work. Within the Complexity results section (lines 260-300), we have also added additional exposition as to what this measure shows, and what we might expect it to look like under different conditions (e.g., where the writer focused edits more narrowly).

---

## [Decision Letter · Decision Letter 2]

16 Feb 2023

PONE-D-22-13798R2Exploitation and exploration in text evolution. Quantifying

planning and translation flows during writing.PLOS ONE

Dear Dr. Lo Sardo,

Thank you for submitting your manuscript to PLOS ONE. After careful consideration, we feel that it has merit but does not fully meet PLOS ONE’s publication criteria as it currently stands. Therefore, we invite you to submit a revised version of the manuscript that addresses the points raised during the review process.

ACADEMIC EDITOR:

Dear authors,

As you can see from the attached comments, both Reviewers consider that your manuscript has really improved as, pending some minor revisions, it is valuable for publication in PlosOne. I particularly appreciated how the result section has been organized, with accurate description of each measure before the corresponding results. Please, solve the minor points raised by Reviewer 1 who asked for few additional methodological clarifications. I am looking forward to receiving your revised manuscript.  

Best regards,

Francesca Peressotti

We look forward to receiving your revised manuscript.

Kind regards,

Francesca Peressotti, Ph.D

Academic Editor

PLOS ONE

Journal Requirements:

Reviewers' comments:

Reviewer's Responses to Questions

**Comments to the Author**

1. If the authors have adequately addressed your comments raised in a previous round of review and you feel that this manuscript is now acceptable for publication, you may indicate that here to bypass the “Comments to the Author” section, enter your conflict of interest statement in the “Confidential to Editor” section, and submit your "Accept" recommendation.

Reviewer #1: (No Response)

Reviewer #2: All comments have been addressed

2. Is the manuscript technically sound, and do the data support the conclusions?

Reviewer #1: Yes

Reviewer #2: Yes

3. Has the statistical analysis been performed appropriately and rigorously? 

Reviewer #1: I Don't Know

Reviewer #2: Yes

4. Have the authors made all data underlying the findings in their manuscript fully available?

Reviewer #1: Yes

Reviewer #2: (No Response)

5. Is the manuscript presented in an intelligible fashion and written in standard English?

Reviewer #1: Yes

Reviewer #2: Yes

6. Review Comments to the Author

Reviewer #1: My name is Mark Torrance, and I reviewed two previous versions of this paper.

First I want to say again that I think the methods described in this paper have the potential to make a important and significant contribution to our understanding of how writers produce extended texts. This is a difficult multi-dimensional problem. The approach is conceptually different from anything I've seen before and I think the academic community needs to hear about it. I therefore would very much like to see this paper published.

Second, I congratulate the authors on their new framing of their methods. As someone very close to, and critical of, the field, there is detail here that I'd disagree with, but nothing that needs revision. I think this is a very competent summary, particularly for a non-specialist audience. I also appreciate the way that this new framing is worked into the rationale for their measures, and interpretation of findings.

I have made a few specific comments below, written as I read through the text. I hope they make sense.

My major remaining concern is still that there are bits of the method that I don't understand. Certainly not at the level at which I could imagine reproducing the analysis, but for some measures (twist ratio in particular), I don't feel I have enough of a grasp of how these were computed to properly understand what is being measured. For a paper that has as a primary focus introducing new ways of characterising how texts develop, I don't think readers should have to take on trust that measures measure what the authors say they measures (and clearly this also isn't the authors' intention). So I am still asking for some clearer explanation in places. Some more detail in my notes below.

Partly with this in mind, I think the authors should consider separating out description of measures - with associated illustrative plots (Figures 1a, 2a, 3) and their reporting of results. In the results section, in addition to what the authors already provide, I would like to see a couple of other things: Some analyses that show the relationship among measures (aggregating within writer/text). Just scatterplots with regression lines would be fine here. So are texts with a high exploration coefficient have higher or lower twist ratios, and so forth. And also relationship between aggregate measures and (a) total writing time (I realise that this is closely related to version index - and you've already done this for the exploration measure) , and (b) final text length (in words).

Specific comments

Abstract: "More than 60 junior researchers" - give exact N.

"A substantial body of theoretical and experimental work across psychology, linguistics, and education has developed well-accepted models of the process of writing. " So this is an over-correction and has swung too far in the other direction. There is a reasonably substantial body of work in this area, and there are some well-cited (though often not well-supported) theories. But understanding of processes underlying written production remains weak, and lags a long way behind, for example, understanding of how people read. I think all you need to say here is just that there is an existing and growing body of psycholinguistic and educational research.

"Here, we propose a new method that can efficiently detect patterns indicative of writing sub-cycles in much smaller datasets than existing NLP methods, presenting a much smaller overhead than previous work such as S-notation [15] given that it does not require keystroke logging to reconstruct the activity of the author." I'm not clear here what overheads the authors are referring to. Collecting keystrokes and then reconstructing all of the author's revisions is not computationally too complex (or at least is a problem that's been solved) and datasets are not huge. Versioning is an alternative approach, and there might be some practical advantages. However, I don't see this as a major contribution of the authors method. Also, given that any version could be reconstructed from a keystroke log, it would presumably be possible to apply them to keystroke data (after some post-processing). My point here is just that the methods that the authors describe contribute something conceptually valuable (and more important) than the fact that text-states were captured by versioning. Versioning isn't a main selling point (or perhaps not a selling point at all).

There is a very short introduction to the key concepts of complexity, exploration, and the twist ratio at the very end of the introduction. I suggest another paragraph, at the start of the analysis / results section of the paper (p6) that unpacks these a bit more and provides orientation to the three sections that follow.

Complexity / Figure 1. I don't know whether this is because this text has been revised, or whether I've finally given it enough attention, but I think I understand the approach here, and what Figure 1 shows. However I'm am still puzzled by something. I was confused by how you handled inserted sentences without changing the index of all subsequent sentences (and therefore seeing them all as having been revised). But then I noticed "taking care to add place-holder elements for the sentences that will be inserted later in the draft so that unmodified sentences will correspond to the same positional index as the draft evolves". Good. But I don't understand how this could be achieved. Or how you handle sentence deletion. So I'm puzzled again about how this works.

Twist Ratio. The explanatory panel in Figure 3, combined with the text, gives me a good understanding, I think, of how you are measuring translation flow, and I think this is a valuable idea. Beyond that I get very lost, however, in particular I don't understand lines 402 to 416. I understand twist ratio measures as angle. But I don't understand "we must decide how many dimensions to use" - dimensions in what space? I'm clearly missing something here, which suggests the need for some more careful unpacking and explanation.

Minor point, but this "If the writer follows this “path of least resistance” very directly, this is an indicator that their original plan for the text went largely unchanged during the process of writing." isn't necessarily the case. Students often write in a very linear way, with very little revision, but this could equally be because they are simply following a train of thought - allowing previous sentences to cue what they say next- and are not particularly motivated to then restructure what they've written. This is the sort of behaviour you would expect if, for example, you asked a student to write a 5 paragraph essay on a familiar topic, in a low-stakes context (or when writing in an exam). In fact, strongly linear writing, with little revision, is, I think ) less likely when writers have a clear plan. Because in this case the text they produce interacts with (is evaluated in light of) their intention. Plans set out communicative intentions for the text but (obviously) not the surface form of the text that is to be produced to fulfill these. Working out these intentions in the final text typically involves trial and error. Hence strong plans tend to result in more revision.

Minor presentational point: I suggest removing contours from Figure 2b and 4c - they don't add anything useful and makes the plot look like it's showing more than it does.

Draft vs. version: The authors use "draft", "version" and "draft version" interchangeably, I think. For me "draft " suggest a version that is identified by the author as significant (ready to show to a co-author, perhaps). Version is a better term for the text that's captured every 1 minute / 3 minutes .

I don't find breakdown by workshop session useful in density plots. I'm not really sure what I should do with this information. Similarly, I think the authors could probably just say "across three workshop sessions (Ns = )" or similar in the methods rather than giving more detail.

Reviewer #2: I commend the authors on the work done in revising the manuscript. I appreciated that they have addressed my major concerns and I am satisfied with the revision. The explanation of their aims and measures is much clearer. Terminology is better specified, and the flow of the manuscript is improved.

I report an oversight: the sentence in lines 85-87 is incomplete ("In analogy with the study of strategies in evolutionary game theory we also use the term exploration to refer to planning phases, where the author tests different options for the draft, an").

7. PLOS authors have the option to publish the peer review history of their article (what does this mean?). If published, this will include your full peer review and any attached files.

Reviewer #1: No

Reviewer #2: **Yes: **Tania Cerni

---

## [Author Response · Author response to Decision Letter 2]

9 Mar 2023

Response to Reviewer #1

Thank you for taking the time to review our paper and for your positive comments regarding the potential significance of our study to the academic community.

We are grateful for your congratulations on the new framing of our methods, and we are pleased to hear that you found the summary to be a competent and comprehensive description. We are glad that you appreciate how we worked the new framing into the rationale for our measures and interpretation of findings.

We thank you for the specific comments you made throughout the text. However, we would like to address your main concern about the method's clarity. 

We considered separating the description and results in a previous stage of the review process but following suggestions from the other reviewer, we ended up in the current structure where each result is preceded by a descriptive introduction. We have however updated the description of our methods, thanks in no small part to your comments, and hope that this further clarification will prove to make the work easier to comprehend and more cohesive.

In the results section, in addition to what the authors already provide, I would like to see a couple of other things: Some analyses that show the relationship among measures (aggregating within writer/text). Just scatterplots with regression lines would be fine here. So are texts with a high exploration coefficient have higher or lower twist ratios, and so forth. And also relationship between aggregate measures and (a) total writing time (I realise that this is closely related to version index - and you've already done this for the exploration measure) , and (b) final text length (in words).

We thank the reviewer for the opportunity to add depth to our work. We have added a table indicating correlations among the metrics and their significance. In order to satisfy the request for scatterplots we have added a supplementary information document that contains scatters of all combinations of metrics and the aggregate measures requested by the reviewer. We are sorry we cannot really provide total writing time as such, since the workshops had diverse structures that cannot always be accounted for. We however provide plots for the relation with version counts, that as the reviewer indicates is strongly related to writing time.

Regarding your specific comments, we will make the necessary revisions to the abstract to give the exact number of junior researchers involved in the study.

“61 junior researchers in science wrote a scientific essay intended for a general readership.”

We appreciate your feedback on the introduction to the paper and have revised it to accurately reflect the existing body of psycholinguistic and educational research.

“A body of theoretical and experimental work across psychology, linguistics, and education has developed well-accepted models of the process of writing.”

"Here, we propose a new method that can efficiently detect patterns indicative of writing sub-cycles in much smaller datasets than existing NLP methods, presenting a much smaller overhead than previous work such as S-notation [15] given that it does not require keystroke logging to reconstruct the activity of the author." I'm not clear here what overheads the authors are referring to. Collecting keystrokes and then reconstructing all of the author's revisions is not computationally too complex (or at least is a problem that's been solved) and datasets are not huge. Versioning is an alternative approach, and there might be some practical advantages. However, I don't see this as a major contribution of the authors method. Also, given that any version could be reconstructed from a keystroke log, it would presumably be possible to apply them to keystroke data (after some post-processing). My point here is just that the methods that the authors describe contribute something conceptually valuable (and more important) than the fact that text-states were captured by versioning. Versioning isn't a main selling point (or perhaps not a selling point at all).

We agree that the use of versioning systems is not a major contribution of our work, we do however hold that the capacity of our methods to be applied to data that is more coarse-grained than keystroke logging, often already available through other systems that make use of versioning, is an advantage with respect to systems that rely exclusively on keystroke logging data. In this sense, we thank you for pointing out the lack of clarity in our work and have rewritten this paragraph in order to better state that the advantages of our methods mainly lie in the possibility of reusing data already available.

“Here, we propose a new method that can efficiently detect patterns indicative of writing sub-cycles in much smaller datasets than existing NLP methods, presenting a much smaller overhead than previous work such as S-notation given that it does not require keystroke logging to reconstruct the activity of the author. Since it relies on automatic versioning that is widespread across many text-editing tools, the measures introduced here can be used to investigate numerous text-based tasks for which data is already available, reducing the overhead in the data gathering process. This kind of data allows us to reconstruct a representation of the editing process which, in turn, allows us to measure its temporal complexity, and indicate when an author is in an exploratory mindset when they are effortlessly translating ideas onto the page, and when they are shifting rapidly between these tasks.”

There is a very short introduction to the key concepts of complexity, exploration, and the twist ratio at the very end of the introduction. I suggest another paragraph, at the start of the analysis / results section of the paper (p6) that unpacks these a bit more and provides orientation to the three sections that follow.

“In the following sections we will describe the analysis of this data divided in three main contributions: a measure of the complexity of the writing process in terms the degree to which it is a sequential process, a measure of the amount of exploratory work that goes into the writing of a text but not into the final version and a measure of the translation flow to exploration ratio in the writing steps.”

Complexity / Figure 1. I don't know whether this is because this text has been revised, or whether I've finally given it enough attention, but I think I understand the approach here, and what Figure 1 shows. However I'm am still puzzled by something. I was confused by how you handled inserted sentences without changing the index of all subsequent sentences (and therefore seeing them all as having been revised). But then I noticed "taking care to add place-holder elements for the sentences that will be inserted later in the draft so that unmodified sentences will correspond to the same positional index as the draft evolves". Good. But I don't understand how this could be achieved. Or how you handle sentence deletion. So I'm puzzled again about how this works.

We understand your confusion about how we handle inserted sentences without changing the index of all subsequent sentences. We hope we have clarified this point by expanding on how we treat deletion. In essence, we handle deleted sentences as empty placeholders so as to keep the alignment of the rest of the work.

“For each subsequent version of a draft after the first, we can define the edit sequence B(t). In order to correctly compare different versions we add place-holder elements (empty strings) for the sentences that will be inserted later in the draft. At the same time we keep deleted sentences as empty place holder elements. This way, unmodified sentences will correspond to the same positional index as the draft evolves.”

Twist Ratio. The explanatory panel in Figure 3, combined with the text, gives me a good understanding, I think, of how you are measuring translation flow, and I think this is a valuable idea. Beyond that I get very lost, however, in particular I don't understand lines 402 to 416. I understand twist ratio measures as angle. But I don't understand "we must decide how many dimensions to use" - dimensions in what space? I'm clearly missing something here, which suggests the need for some more careful unpacking and explanation.

For the Twist Ratio section, we have revised the text to explain the selection of the number of dimensions of the embedding space.

“This procedure is possible since edit distance is a well-defined metric. With the set of pairwise distances we can embed the set of versions in an euclidean space. We must, however, determine the number of dimensions to use. High dimensional spaces will guarantee that all distances between versions can be preserved, but much of the structure is lost.”

Minor point, but this "If the writer follows this “path of least resistance” very directly, this is an indicator that their original plan for the text went largely unchanged during the process of writing." isn't necessarily the case. Students often write in a very linear way, with very little revision, but this could equally be because they are simply following a train of thought - allowing previous sentences to cue what they say next- and are not particularly motivated to then restructure what they've written. This is the sort of behaviour you would expect if, for example, you asked a student to write a 5 paragraph essay on a familiar topic, in a low-stakes context (or when writing in an exam). In fact, strongly linear writing, with little revision, is, I think ) less likely when writers have a clear plan. Because in this case the text they produce interacts with (is evaluated in light of) their intention. Plans set out communicative intentions for the text but (obviously) not the surface form of the text that is to be produced to fulfill these. Working out these intentions in the final text typically involves trial and error. Hence strong plans tend to result in more revision.

We thank the reviewer for pointing out that our outlining of the implications of revisions is incomplete. We have edited this paragraph to specify that the broader point we make here is that we do not observe major revisions.

“If the writer follows this “path of least resistance'” very directly, this is an indicator that their original plan for the text went largely unchanged during the process of writing, or that they followed a single line of thought, and in general that they did not commit to major revisions.”

Draft vs. version: The authors use "draft", "version" and "draft version" interchangeably, I think. For me "draft " suggest a version that is identified by the author as significant (ready to show to a co-author, perhaps). Version is a better term for the text that's captured every 1 minute / 3 minutes .

We thank the reviewer for pointing out the lack of clarity on the wording in our work. We have changed all references to versions of a draft as such. We now use the word “draft” only to signify the final version of the work to get to it (such as in the case of “evolution of the draft”).

Response to Reviewr #2

Reviewer #2: I commend the authors on the work done in revising the manuscript. I appreciated that they have addressed my major concerns and I am satisfied with the revision. The explanation of their aims and measures is much clearer. Terminology is better specified, and the flow of the manuscript is improved.

I report an oversight: the sentence in lines 85-87 is incomplete ("In analogy with the study of strategies in evolutionary game theory we also use the term exploration to refer to planning phases, where the author tests different options for the draft, an").

We thank the reviewer for their commendation and for pointing out the oversight. We have corrected it as follows

“In analogy with the study of strategies in evolutionary game theory we also use the term exploration to refer to planning phases, where the author tests different options for the draft.”

---

## [Editor Report · Decision Letter 3]

13 Mar 2023

Exploitation and exploration in text evolution. Quantifying

planning and translation flows during writing.

PONE-D-22-13798R3

Dear Dr. Lo Sardo,

We’re pleased to inform you that your manuscript has been judged scientifically suitable for publication and will be formally accepted for publication once it meets all outstanding technical requirements.

Kind regards,

Francesca Peressotti, Ph.D

Academic Editor

PLOS ONE
---

## [Editor Report · Acceptance letter]

20 Mar 2023

PONE-D-22-13798R3 

Exploitation and exploration in text evolution. Quantifying planning and translation flows during writing. 

Dear Dr. Lo Sardo:

I'm pleased to inform you that your manuscript has been deemed suitable for publication in PLOS ONE. Congratulations! Your manuscript is now with our production department. 

Kind regards, 

on behalf of

Dr. Francesca Peressotti 

Academic Editor

PLOS ONE